# Index2Sort: Sorting Algorithm Using Static Index Structure

**Atsuki Sato**                                                                 *a_sato@hal.t.u-tokyo.ac.jp*
*Graduate School of Information Science and Technology*
*The University of Tokyo*

**Yusuke Matsui**                                                              *matsui@hal.t.u-tokyo.ac.jp*
*Graduate School of Information Science and Technology*
*The University of Tokyo*

**Reviewed on OpenReview:** *https://openreview.net/forum?id=YUmVxjMhpm*

## Abstract

We introduce Index2Sort, a general framework for deriving sorting algorithms from static indexes. Index2Sort treats the index as an opaque box that exposes only two operations: index construction and rank queries. This abstraction allows Index2Sort to be applied to various index structures, including classical and learned indexes. Our theoretical analysis shows that the computational guarantees of the index transfer directly to Index2Sort. If the index can be constructed in expected time $\mathcal{O}(nC(n))$ and can answer rank queries in expected time $\mathcal{O}(Q(n))$, then Index2Sort sorts the input in expected time $\mathcal{O}(nC(n) + nQ(n))$. In particular, when using a state-of-the-art learned index with $C(n) = Q(n) = 1$, this yields an expected complexity of $\mathcal{O}(n)$, which is a strictly tighter bound than those of existing learned sorting algorithms. In contrast to recent theoretical works on learned sorting, which derive complexity guarantees by analyzing the internal structure of a learned index and designing a sorting algorithm with a similar structure, Index2Sort achieves stronger guarantees without requiring any inspection or modification of the index internals.

## 1 Introduction

Recent research integrating machine learning into classical data structures and algorithms has led to dramatic performance improvements in fundamental computational tasks, including indexing and sorting. This line of work has given rise to a new class of algorithms known as *learned indexes* (Kraska et al., 2018) and *learned sorts* (Kraska et al., 2019). Both share a common design principle: they approximate the cumulative distribution function (CDF) of the data with a machine learning model and leverage its predictions within the algorithm. In these areas, researchers have not only demonstrated significant empirical speedups but also developed algorithms with strong expected-time complexity guarantees under distributional assumptions (Croquevielle et al., 2025; Sato & Matsui, 2025; Zeighami & Shahabi, 2024).

There is a noticeable gap between the theoretical progress on learned indexes and learned sorts. Advances in learned sort have historically followed those in learned index, but with some delay. For example, after the development of learned indexes with expected construction time $\mathcal{O}(n \log \log n)$ and expected rank query time $\mathcal{O}(\log \log n)$ (Zeighami & Shahabi, 2023), researchers carefully examined the internal structure of such learned indexes and redesigned them for sorting, eventually producing learned sorts with expected $\mathcal{O}(n \log \log n)$ time (Sato & Matsui, 2025; Zeighami & Shahabi, 2024). Later, learned indexes improved further to expected $\mathcal{O}(n)$-time construction and $\mathcal{O}(1)$-time rank queries under similar assumptions (Croquevielle et al., 2025). Nevertheless, the best existing learned sorts still have expected $\mathcal{O}(n \log \log n)$ time.

This situation naturally raises two questions: (1) Under the same assumptions as (Croquevielle et al., 2025), can we design a learned sort with expected $\mathcal{O}(n)$ time? (2) If even stronger learned indexes are developed in the future, can their improvements be automatically transferred to learned sorts?

Our answer to both questions is yes. In this work, we present **Index2Sort**, the first general framework that derives sorting algorithms from any static index. The algorithmic structure of Index2Sort is intentionally simple and resembles classical recursive bucketing and samplesort-style pipelines. Our contribution is not a new bucketing mechanism itself, but rather the opaque-box abstraction and its analysis: Index2Sort treats the index only through construction and rank-query operations, and automatically transfers their computational guarantees to sorting.

This abstraction allows Index2Sort to automatically inherit the computational guarantees of the underlying index, thereby bridging the theoretical gap between learned indexes and learned sorts. Specifically, if the index can be constructed in expected time $\mathcal{O}(nC(n))$ and answer rank queries in expected time $\mathcal{O}(Q(n))$, then Index2Sort sorts the input in expected time $\mathcal{O}(nC(n) + nQ(n))$. As a concrete example, applying the state-of-the-art learned index of Croquevielle et al. (2025) immediately yields a sorting algorithm with expected running time $\mathcal{O}(n)$ under standard distributional assumptions. Furthermore, thanks to the generality of Index2Sort, if learned indexes with even stronger theoretical guarantees are developed in the future, their benefits will carry over directly to sorting.

Our contributions are summarized as follows:

- **General opaque-box sorting framework**: We propose Index2Sort, the first framework that performs sorting by treating any static index as an opaque box. This achieves a conceptual inversion of the usual dependency between indexes and sorting: although an index is constructed over a sorted array, we demonstrate that it can itself be used for sorting.

- **Automatic inheritance of guarantees**: We formally prove that Index2Sort automatically inherits the computational guarantees of the underlying index, thereby establishing a formal and general theoretical bridge between indexing and sorting.

- **State-of-the-art theoretical guarantees for sorting**: By instantiating Index2Sort with state-of-the-art learned indexes, we immediately obtain algorithms that achieve expected running time $\mathcal{O}(n)$ under the standard distributional assumptions, and we further show that $\mathcal{O}(n \log \log n)$ can still be achieved under even weaker assumptions. These results are provably stronger complexity guarantees than all existing learned sorting algorithms.

- **Future-proof paradigm**: Beyond these results, Index2Sort offers a paradigm that continuously benefits from progress in index research: any theoretical advance in indexing immediately translates into an advance in sorting.

This paper is organized as follows. Section 2 discusses related work. Section 3 introduces the necessary definitions and notation, and Section 4 presents Index2Sort and its complexity guarantees. Section 5 validates these guarantees experimentally. Section 6 examines limitations, and Section 7 concludes the paper.

## 2 Related Work

Here, we first give an overview of indexes and sorting methods, focusing on learned indexes and learned sorts in Section 2.1. Then, in Section 2.2, we introduce algorithms with predictions, a closely related field, and discuss its connections and differences with our work.

### 2.1 Learned Index and Learned Sort

An index, in a broader sense, is a data structure for fast data access. Examples include B-trees (Bayer & McCreight, 1972), hash maps (Knuth, 1998), and Bloom filters (Bloom, 1970), which are widely used in applications such as databases (Ramakrishnan & Gehrke, 2002), search engines (Schütze et al., 2008), and file systems (Ghemawat et al., 2003). Recently, *learned indexes* have been proposed (Kraska et al., 2018), replacing or augmenting classical structures with machine learning models to improve memory efficiency and query speed. Research has explored machine learning-augmented versions of various data structures, including Bloom filters (Mitzenmacher, 2018; Dai & Shrivastava, 2020; Vaidya et al., 2021; Sato & Matsui,

2023), R-trees (Gu et al., 2023; Abdullah-Al-Mamun et al., 2022; Hidaka & Matsui, 2024; Tang et al., 2026a;b), and count-min sketches (Hsu et al., 2019; Zhang et al., 2020; Dolera et al., 2023; Nishishita et al., 2025). In particular, learned indexes with functionality similar to B-tree have been extensively studied (Galakatos et al., 2019; Ferragina & Vinciguerra, 2020; Sun et al., 2023) and are often referred to as learned indexes in the narrow sense. These approaches use machine learning models to approximate the CDF of the input array distribution, enabling better memory efficiency and faster search. Most learned indexes employ a hierarchical structure of linear models (Galakatos et al., 2019; Ferragina & Vinciguerra, 2020; Ding et al., 2020; Wang et al., 2020; Hadian & Heinis, 2020; Li et al., 2021; Sato et al., 2026), though other designs, such as those based on polynomial functions (Wu et al., 2021) or neural networks (Kraska et al., 2018), have also been proposed. More recently, there has been increasing interest in learned indexes with theoretical guarantees. Details of these guarantees are discussed in Section 4.3.

Sorting is a fundamental problem in computer science, and many algorithms have been proposed. Any comparison-based sorting algorithm requires $\Omega(n \log n)$ comparisons in the worst case; for example, Mergesort matches this bound. With additional assumptions or side information, it is possible to achieve lower worst-case complexity. For instance, RadixSort achieves a worst-case complexity of $\mathcal{O}(nw)$, where $w$ is the number of digits per element. For integer arrays, deterministic algorithms with a complexity of $\mathcal{O}(n \log \log n)$ (Han, 2002) and randomized algorithms with an expected complexity of $\mathcal{O}(n\sqrt{\log \log n})$ (Han & Thorup, 2002) have been proposed. For real-valued arrays, a recent algorithm achieves a complexity of $\mathcal{O}(n\sqrt{\log n})$ (Han, 2020). Inspired by learned indexes, sorting algorithms using machine learning models to approximate the CDF, referred to as *learned sort*, have been proposed (Kraska et al., 2019). The learned sort algorithms perform sorting quickly by efficiently assigning keys to buckets using the predicted CDF and reducing comparisons (Kristo et al., 2020; 2021). More recently, by redesigning the architecture of learned indexes for sorting, learned sorting algorithms with $\mathcal{O}(n \log \log n)$ expected complexity have been introduced under bounded-density i.i.d. input assumptions (Sato & Matsui, 2025; Zeighami & Shahabi, 2024), which are formally defined in Section 3. In contrast, our Index2Sort adopts a different approach by treating the index as an opaque box, achieving stronger guarantees.

## 2.2 Algorithms with predictions

Algorithms with predictions (Mitzenmacher & Vassilvitskii, 2022) are a rapidly growing research field, demonstrating that incorporating predictions can substantially improve the performance of certain algorithms. Early research in this area focused primarily on classic online problems, such as caching (Narayanan et al., 2018; Rohatgi, 2020; Lykouris & Vassilvitskii, 2021; Antoniadis et al., 2023b; Sadek & Elias, 2024), rent-or-buy problems (Purohit et al., 2018; Gollapudi & Panigrahi, 2019; Shin et al., 2023), and scheduling (Mitzenmacher, 2020; Lattanzi et al., 2020; Lassota et al., 2023; Elias et al., 2024). The scope of these techniques has been extended to offline problems, including matching (Dinitz et al., 2021; Sakaue & Oki, 2022; Choo et al., 2024), clustering (Ergun et al., 2022; Nguyen et al., 2023), and graph algorithms (Chen et al., 2022; Davies et al., 2023; Polak & Zub, 2024). There has also been significant progress in sorting with predictions (Lu et al., 2021; Chan et al., 2023; Erlebach et al., 2023). In particular, (Bai & Coester, 2023) proposed a generalized sorting algorithm with predictions that offer tight complexity guarantees.

While many studies assume that predictions are passively obtained at no cost, others focus on optimizing the predictions themselves. For example, there are studies that propose algorithms to reduce the number of predictions used (Im et al., 2022; Drygala et al., 2023; Benomar & Perchet, 2023; Aamand et al., 2023; Sadek & Elias, 2024), and some limit the size per prediction (Mitzenmacher, 2021; Dütting et al., 2021; Antoniadis et al., 2023a). In addition, research efforts have been made to design customized loss functions for training machine learning models used to generate predictions (Du et al., 2021; Anand et al., 2020) or to train machine learning models dynamically using online-learning methods (Khodak et al., 2022; Sakaue & Oki, 2022; 2023). These studies share a common direction in that they refine the predictions themselves.

While our work shares similarities with these approaches, it differs in that we treat the training time of the machine learning model as part of the computational cost, explicitly accounting for both training and inference. This perspective is particularly crucial for end-to-end performance analysis in offline problems, where predictions are tailored to each individual instance.

## 3 Preliminaries

Here, we introduce several definitions and notations required for our problem setup.

**Sorting.** Sorting is the operation that converts an input array into a sorted array. Let $\boldsymbol{x} = [x_1, x_2, \ldots, x_n]$ be an array of $n$ real numbers. The array $\boldsymbol{x}$ may contain duplicate elements; in other words, there may exist indices $i, j$ such that $x_i = x_j$. Sorting transforms $\boldsymbol{x}$ into $\boldsymbol{x}' = [x_{\pi(1)}, x_{\pi(2)}, \ldots, x_{\pi(n)}]$, where $\pi$ is a bijective function from $\{1, 2, \ldots, n\}$ to $\{1, 2, \ldots, n\}$ that satisfies $x'_i \leq x'_j$ for all $i, j$ such that $i < j$. In this paper, we use a prime symbol ($'$) on a vector (e.g., $\boldsymbol{x}'$) to denote its sorted version.

**Indexing.** Algorithms for static index data structures consist of two phases: the construction phase and the query response phase. In the construction phase, a sorted array $\boldsymbol{x}' \in \mathbb{R}^n$ is provided, and an index data structure is constructed. The constructed index does not necessarily support the insertion or deletion of elements. In the query response phase, the index data structure processes a given query $q \in \mathbb{R}$ and returns the rank of $q$, that is, the number of elements in the array $\boldsymbol{x}'$ that are less than $q$. We assume only the above two functionalities of the index and make no other assumptions, such as the internal structure.

**Data Distribution and Distribution Shift.** The theoretical guarantees for learned indexes and learned sorts often rely on distributional assumptions. For completeness, we recall representative distribution classes used in prior theoretical studies of learned indexes and learned sorts (Zeighami & Shahabi, 2023; Sato & Matsui, 2025; Zeighami & Shahabi, 2024; Croquevielle et al., 2025). However, note that the theoretical analysis of Index2Sort does not rely on the specific details of these distribution classes; as discussed in Section 4.2, given an index whose construction and query costs are guaranteed under a certain distributional assumption, Index2Sort automatically yields a sorting algorithm with corresponding guarantees under the same assumption. The transfer theorem and its proof are agnostic to the particular distribution class.

We define an array $\boldsymbol{D} = [D_1, D_2, \ldots, D_n]$ as being *sampled independently from distributions* $\boldsymbol{\chi} = [\chi_1, \chi_2, \ldots, \chi_n]$ if each $D_i$ is drawn independently from $\chi_i$ for all $i = 1, 2, \ldots, n$. For brevity, we write this as $\boldsymbol{D} \sim \boldsymbol{\chi}$. When all elements of $\boldsymbol{D}$ are sampled i.i.d. from a single distribution $\chi$, we denote this as $\boldsymbol{D} \overset{\text{iid}}{\sim} \chi$. If $\boldsymbol{D} \sim \boldsymbol{\chi}$ and $\chi_i \in \mathfrak{X}$ for all $i$, we state that $\boldsymbol{D}$ is *sampled from the distribution class* $\mathfrak{X}$, where $\mathfrak{X}$ is a set of distributions. We define the following representative distribution classes:

- $\mathfrak{X}_{\rho_1, \rho_2, \mathcal{K}}$ ($0 < \rho_1 \leq \rho_2 < \infty$, $\mathcal{K}$ is a finite continuous domain): The set of distributions with probability density functions $f$ over $\mathcal{K}$ such that $\forall x \in \mathcal{K}, \rho_1 \leq f(x) \leq \rho_2$.

- $\mathfrak{X}_{\rho_f, \mathcal{K}}$ ($0 < \rho_f < \infty$, $\mathcal{K}$ is a finite continuous domain): The set of distributions with probability density functions $f$ over $\mathcal{K}$ such that $\int_{\mathcal{K}} f^2(x) dx \leq \rho_f$.

- $\mathfrak{X}_C$ ($C > 0$): The set of subexponential distributions with the tail decay parameter $C$. Formally, if $X \sim \chi$ for some $\chi \in \mathfrak{X}_C$, then $\Pr[|X| \geq x] \leq 2e^{-Cx}$ for all $x \geq 0$.

The class $\mathfrak{X}_{\rho_1, \rho_2, \mathcal{K}}$ appears in (Zeighami & Shahabi, 2023; Sato & Matsui, 2025; Zeighami & Shahabi, 2024), while $\mathfrak{X}_{\rho_f, \mathcal{K}}$ and $\mathfrak{X}_C$ are used in (Croquevielle et al., 2025). The distribution class $\mathfrak{X}_{\rho_f, \mathcal{K}}$ is broader than $\mathfrak{X}_{\rho_1, \rho_2, \mathcal{K}}$, and $\mathfrak{X}_C$ is broader than $\mathfrak{X}_{\rho_f, \mathcal{K}}$. More precisely, the following holds:

> **Proposition 3.1.** *For any $\rho_1$, $\rho_2$, and $\mathcal{K}$, there exists a constant $\rho_f$ such that $\mathfrak{X}_{\rho_1, \rho_2, \mathcal{K}} \subseteq \mathfrak{X}_{\rho_f, \mathcal{K}}$. Furthermore, for any $\rho_f$ and $\mathcal{K}$, there exists a constant $C$ such that $\mathfrak{X}_{\rho_f, \mathcal{K}} \subseteq \mathfrak{X}_C$.*

*Proof.* Let $\chi \in \mathfrak{X}_{\rho_1, \rho_2, \mathcal{K}}$ with density $f$ on a finite domain $\mathcal{K}$. Since $f(x) \leq \rho_2$ for all $x \in \mathcal{K}$, we have $\int_{\mathcal{K}} f^2(x) \, dx \leq \rho_2 \int_{\mathcal{K}} f(x) \, dx = \rho_2$. Therefore, with $\rho_f = \rho_2$, we have $\chi \in \mathfrak{X}_{\rho_f, \mathcal{K}}$, implying $\mathfrak{X}_{\rho_1, \rho_2, \mathcal{K}} \subseteq \mathfrak{X}_{\rho_f, \mathcal{K}}$.

Next, let $\chi \in \mathfrak{X}_{\rho_f, \mathcal{K}}$ and $X \sim \chi$. Since $\mathcal{K}$ is a continuous finite domain, let $B = \sup\{|x| \mid x \in \mathcal{K}\} < \infty$. Then $\Pr[|X| \geq x] = 0$ for all $x > B$. Setting $C = (\ln 2)/B$, we have $2e^{-Cx} \geq 1$ for all $0 \leq x \leq B$, and hence $\Pr[|X| \geq x] \leq 2e^{-Cx}$ for all $x \geq 0$. Therefore $\chi \in \mathfrak{X}_C$, implying $\mathfrak{X}_{\rho_f, \mathcal{K}} \subseteq \mathfrak{X}_C$. $\qquad\square$

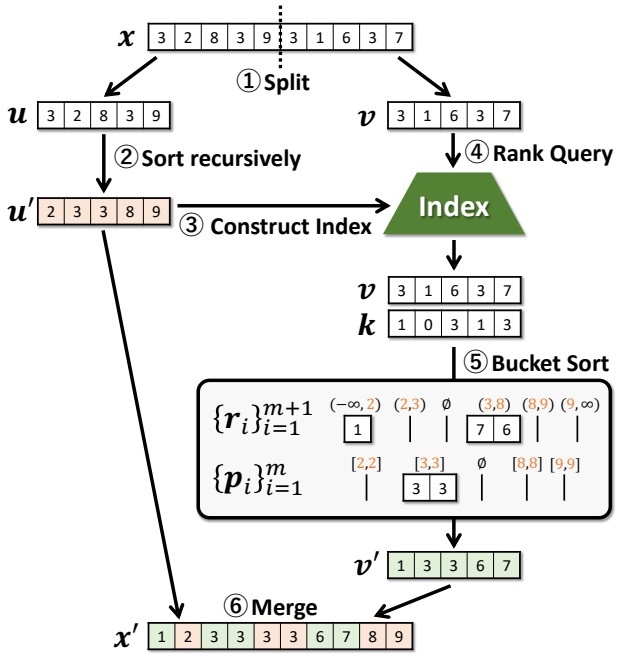

Figure 1: Index2Sort first recursively sorts part of the input, builds an index over it, and then uses this index to bucket-sort the remaining elements.

**Algorithm 1** Index2Sort

1: **Input:** $\boldsymbol{x} \in \mathbb{R}^n$ (the array to be sorted)
2: **Output:** $\boldsymbol{x}' \in \mathbb{R}^n$ (the sorted version of $\boldsymbol{x}$)
3: **function** INDEX2SORT($\boldsymbol{x}$)
4:     $n \leftarrow |\boldsymbol{x}|, \quad m \leftarrow \lfloor n/2 \rfloor$
5:     **if** $n < \tau$ **then**
6:         **return** MERGESORT($\boldsymbol{x}$)
7:     $\boldsymbol{u} \leftarrow \boldsymbol{x}[1:m], \quad \boldsymbol{v} \leftarrow \boldsymbol{x}[m+1:n]$     ⟩ ①
8:     $\boldsymbol{u}' \leftarrow$ INDEX2SORT($\boldsymbol{u}$)     ⟩ ②
9:     $\mathcal{I} \leftarrow$ CONSTRUCTINDEX($\boldsymbol{u}'$)     ⟩ ③
10:    $\boldsymbol{k} \leftarrow [\,]$
11:    **for** $i = 1, \ldots, n - m$     ⟩ ④
12:        $\boldsymbol{k}$.append($\mathcal{I}$.rank($v_i$))
13:    $\boldsymbol{r}_1 \leftarrow [\,], \ldots, \boldsymbol{r}_{m+1} \leftarrow [\,]$
14:    $\boldsymbol{p}_1 \leftarrow [\,], \ldots, \boldsymbol{p}_m \leftarrow [\,]$
15:    **for** $i = 1, \ldots, n - m$
16:        **if** $k_i = m \lor v_i \neq u'_{k_i+1}$ **then**
17:            $\boldsymbol{r}_{k_i+1}$.append($v_i$)
18:        **else**     ⟩ ⑤
19:            $\boldsymbol{p}_{k_i+1}$.append($v_i$)
20:    **for** $i = 1, \ldots, m + 1$
21:        $\boldsymbol{r}'_i \leftarrow$ MERGESORT($\boldsymbol{r}_i$)
22:    $\boldsymbol{v}' \leftarrow$ CONCAT($\boldsymbol{r}'_1, \boldsymbol{p}_1, \ldots, \boldsymbol{p}_m, \boldsymbol{r}'_{m+1}$)
23:    $\boldsymbol{x}' \leftarrow$ MERGE($\boldsymbol{u}', \boldsymbol{v}'$)     ⟩ ⑥
24:    **return** $\boldsymbol{x}'$

To quantify distribution shift, we use total variation distance as in (Zeighami & Shahabi, 2024). For a sequence of distributions $\boldsymbol{\chi} = [\chi_1, \chi_2, \ldots, \chi_n]$, define $\Delta(\boldsymbol{\chi}) = \max_{\chi_i, \chi_j \in \boldsymbol{\chi}} d_{\mathrm{TV}}(\chi_i, \chi_j)$, where $d_{\mathrm{TV}}(\chi_i, \chi_j)$ represents the total variation distance between $\chi_i$ and $\chi_j$. The value of $\Delta(\boldsymbol{\chi})$ lies between 0 and 1, with $\Delta(\boldsymbol{\chi}) = 0$ indicating that all distributions in $\boldsymbol{\chi}$ are identical.

# 4 Method: Index2Sort

In this section, we first describe the proposed Index2Sort algorithm in Section 4.1, then present its complexity theorems in Section 4.2, and finally summarize in Section 4.3 the corollaries obtained by applying our framework to several known indexes.

## 4.1 Algorithm of Index2Sort

Index2Sort recursively sorts a portion of the input array, constructs an index using the sorted portion, and then performs bucket sort on the remaining elements of the input array using the constructed index. The algorithm is visualized in Figure 1 and its pseudocode is presented in Algorithm 1. If the length of the input array is smaller than a certain threshold $\tau$ ($\geq 1$), we sort the array using a standard algorithm, such as MergeSort. In the following, let the length of the input array be $n$ ($\geq \tau$) and the input array be $\boldsymbol{x}$ ($\in \mathbb{R}^n$).

First, the input array $\boldsymbol{x}$ is split into two parts, $\boldsymbol{u}$ and $\boldsymbol{v}$. For theoretical guarantees, $\boldsymbol{x}$ is shuffled once using an $\mathcal{O}(n)$ algorithm, such as the Fisher–Yates shuffle (Fisher & Yates, 1953), before being split into $\boldsymbol{u}$ and $\boldsymbol{v}$. This shuffle is performed only once and is not required during subsequent recursive calls. We define $\boldsymbol{u} = \boldsymbol{x}[1:m]$ and $\boldsymbol{v} = \boldsymbol{x}[m+1:n]$, where $m = \lfloor \alpha n \rfloor$ for an arbitrary constant $\alpha \in (0, 1)$. For simplicity, we assume $\alpha = 1/2$, i.e., $m = \lfloor n/2 \rfloor$, in the following explanation. However, the algorithm and its computational guarantees remain valid for any $\alpha \in (0, 1)$.

The algorithm then recursively sorts $\boldsymbol{u}$ using Index2Sort. Since the length of $\boldsymbol{u}$ is strictly smaller than that of $\boldsymbol{x}$, this recursion must eventually terminate and does not lead to an infinite loop. After obtaining the

sorted array $\boldsymbol{u}'$, an index is constructed on $\boldsymbol{u}'$. Note that constructing the index requires a sorted array, and $\boldsymbol{u}'$ satisfies this condition. In this way, Index2Sort makes it possible to utilize static indexes for sorting.

Next, the constructed index is used to bucket-sort $\boldsymbol{v}$. For each $v_i \in \boldsymbol{v}$, we perform a rank query on the index, obtaining $\boldsymbol{k} \in \{0, \ldots, m\}^{n-m}$ where $k_i$ is the rank of $v_i$ in $\boldsymbol{u}'$. We then prepare $m+1$ range buckets $(\boldsymbol{r}_1, \ldots, \boldsymbol{r}_{m+1})$ and $m$ point buckets $(\boldsymbol{p}_1, \ldots, \boldsymbol{p}_m)$ (as detailed in Section 4.2, we introduce these two types of buckets for theoretical guarantees). Each range bucket stores elements that fall within the open intervals between successive elements of $\boldsymbol{u}'$, while each point bucket stores values that exactly match certain elements of $\boldsymbol{u}'$. Concretely, $v_i$ is placed into $\boldsymbol{r}_{k_i+1}$ if $k_i = m$ or $v_i \neq u'_{k_i+1}$; otherwise, it is placed into $\boldsymbol{p}_{k_i+1}$.

Each range bucket is then sorted. We require the range-bucket sorter to run in $\mathcal{O}(n^2)$ time for theoretical guarantees; this is discussed in detail in Section 4.2 and Appendix A. In our implementation, however, we use IntroSort (Musser, 1997), which has $\mathcal{O}(n \log n)$ worst-case time. The range and point buckets are then merged alternately to produce the sorted array $\boldsymbol{v}'$. Finally, $\boldsymbol{u}'$ and $\boldsymbol{v}'$ are merged in the manner of MergeSort to produce the array $\boldsymbol{x}'$, which is the sorted version of $\boldsymbol{x}$. For the final merging step, one could instead obtain the same sorted output by concatenating $\boldsymbol{u}'$, the point buckets, and the range buckets in the appropriate order. In this paper, we adopt the merge-based description for simplicity of exposition.

## 4.2 Theorems on Complexity of Index2Sort

**Fundamental Theorem.** First, we present the most fundamental and intuitive result, applicable when the complexity guarantees of the index do not rely on distributional assumptions.

> **Theorem 4.1.** *Consider a static index algorithm satisfying: (1) given a sorted array of length $n$, the index is constructed in $\mathcal{O}(nC(n))$ expected time; (2) the index answers a rank query in $\mathcal{O}(Q(n))$ expected time. Then, Index2Sort sorts an array of length $n$ in $\mathcal{O}(nC(n) + nQ(n))$ expected time.*

A rigorous proof is given in Appendix A.1; we outline the intuition here. Note that in the following analysis, we expand the recursion performed in the step ② and accumulate the time complexity for each step from ① to ⑥. Steps ① (splitting) and ⑥ (merging) each take $\mathcal{O}(n)$ time. Step ③ constructs indexes for arrays of lengths $n/2$, $n/4$, … with costs $\mathcal{O}((n/2)C(n/2))$, $\mathcal{O}((n/4)C(n/4))$, …, summing to $\mathcal{O}(nC(n))$ since $C$ is non-decreasing. Similarly, the total complexity of ④ is $\mathcal{O}(nQ(n))$. Therefore, the only nontrivial part is the total expected time complexity of ⑤. We show that this complexity is $\mathcal{O}(n)$ by adapting a classical probabilistic analysis of bucket size distributions in (Frazer & McKellar, 1970) to our setting. This statement relies on the setting $m = \Theta(n)$, equivalently, on using a fixed constant $\alpha \in (0, 1)$. In this regime, the average expected squared bucket size is constant, and hence $\mathbb{E}[\sum_{i=1}^{m+1} A_i^2] = \mathcal{O}(n)$, where $A_i$ is the size of the $i$-th range bucket. Thus, even a quadratic-time algorithm for sorting individual range buckets is sufficient to achieve expected linear time for step ⑤. Therefore, the total time complexity of Index2Sort is $\mathcal{O}(nC(n) + nQ(n))$.

We emphasize that point buckets are essential for Index2Sort to achieve the overall complexity of $\mathcal{O}(nC(n) + nQ(n))$. Without them, simply assigning elements to $m+1$ buckets based on rank queries does not guarantee that step ⑤ runs in $\mathcal{O}(n)$ expected time. For example, if a particular value appears $\Omega(n)$ times in the input array $\boldsymbol{x}$, all occurrences fall into the same bucket, requiring $\Omega(n \log n)$ time to sort the bucket. Index2Sort avoids this by using point buckets: for each element, we perform a constant-time check to decide whether it belongs to a range bucket or a point bucket. This ensures that the expected sum of squared range-bucket sizes is $\mathcal{O}(n)$, and thus sorting all range buckets takes $\mathcal{O}(n)$ expected time. Since all elements in a point bucket are identical and need not be sorted, the total expected cost of step ⑤ remains $\mathcal{O}(n)$. Consequently, Index2Sort preserves the overall $\mathcal{O}(nC(n) + nQ(n))$ complexity even in the presence of many duplicate elements.

We also note that the same guarantee on Index2Sort holds even when the query cost is given in *amortized* expected time. Specifically, suppose that for any $k \geq 1$ rank queries on an index built over an array of length $n$, the expected total query time is $\mathcal{O}(kQ(n))$. Then, the expected running time of Index2Sort remains $\mathcal{O}(nC(n) + nQ(n))$. This is because our analysis uses only the fact that the total expected time spent on rank queries is $\mathcal{O}(nQ(n))$, rather than requiring each individual rank query to take $\mathcal{O}(Q(n))$ expected time.

**Under Distributional Assumptions.** Theorem 4.1 cannot be applied directly when the theoretical guarantee of the index relies on assumptions about the distribution of input arrays and queries, which is common in learned indexes. To cover these cases, we provide two companion results: the i.i.d. setting (Theorem 4.2) and the distribution-shift setting (Theorem 4.3).

> **Theorem 4.2.** *Consider a static index algorithm satisfying: (1) given a sorted array of length $n$ whose elements are sampled i.i.d. from a distribution $\chi \in \mathfrak{X}$, the index is constructed in $\mathcal{O}(nC(n))$ expected time; (2) given a query independently sampled from the same distribution $\chi$, the index returns the rank of the query in $\mathcal{O}(Q(n))$ expected time. Then, Index2Sort sorts an array of $n$ i.i.d. samples from $\chi \in \mathfrak{X}$ in $\mathcal{O}(nC(n) + nQ(n))$ expected time.*

The proof of this theorem follows almost the same steps as the proof of Theorem 4.1, as detailed in Appendix A.1. It is worth noting that the assumption that each element of $\boldsymbol{x}$ is sampled i.i.d. from the distribution $\chi$ propagates to the elements of $\boldsymbol{u}$ and $\boldsymbol{v}$. This propagation ensures that the complexity of constructing the index on $\boldsymbol{u}'$ is bounded by $\mathcal{O}(nC(n))$ and that the complexity of performing rank queries on all elements of $\boldsymbol{v}$ is bounded by $\mathcal{O}(nQ(n))$.

> **Theorem 4.3.** *Consider a static index algorithm satisfying: (1) given a sorted array of $n$ samples from a distribution in $\mathfrak{X}$ with shift at most $\delta$, the index is constructed in $\mathcal{O}(nC(n,\delta))$ expected time; (2) given a query from the same distribution class with shift at most $\delta$, the index returns its rank in $\mathcal{O}(Q(n,\delta))$ expected time. Then, Index2Sort sorts an array sampled from $\mathfrak{X}$ (with $\delta$ distribution shift) in $\mathcal{O}(nC(n,\delta) + nQ(n,\delta))$ expected time.*

The proof of this theorem is similar to that of Theorem 4.2, with the detailed proof given in Appendix A.1. Notably, in the expected time complexity of Index2Sort, $\delta$ appears only in the functions $C$ and $Q$. In other words, the distribution shift impacts only the index construction and query processing steps; the efficiency of the rest of the components of the Index2Sort algorithm is unaffected.

**Handling Approximate Rank Queries.** Furthermore, when the index algorithm supports *approximate rank queries*, which return approximate ranks with a maximum error of $\varepsilon$ instead of exact ranks, the time complexity of Index2Sort can still be guaranteed. This scenario is common because many index structures incorporate mechanisms that provide approximate ranks. These indexes typically refine it to obtain the exact rank through methods such as binary search or exponential search. For example, in a B-tree, each node typically corresponds to a page block that stores multiple data records. As a result, the pure query response of a B-tree has an error bounded by the block size. Similarly, in some learned indexes, such as the PGM-index (Ferragina & Vinciguerra, 2020), the maximum error is explicitly specified as a parameter during the index construction.

The complexity guarantee of Index2Sort under this setting is achieved by making one of the following minor modifications to the algorithm for ⑤: (i) using the sorting algorithm with predictions proposed in (Bai & Coester, 2023) (with slight modifications) to sort $\boldsymbol{v}$ instead of performing bucket sort, or (ii) performing an exponential search on $\boldsymbol{u}'$, starting from the approximate rank to obtain the exact rank. With either modification, the time complexity of Index2Sort can be bounded as follows:

> **Theorem 4.4.** *Consider a static index algorithm satisfying: (1) given a sorted array of length $n$, the index is constructed in $\mathcal{O}(nC(n))$ expected time; (2) the index returns an approximate rank with error at most $\varepsilon$ in $\mathcal{O}(Q(n))$ expected time. If the step ⑤ is implemented using either (i) sorting algorithm with predictions (Bai & Coester, 2023), or (ii) exponential search starting from the approximate rank, then Index2Sort sorts in $\mathcal{O}(nC(n) + nQ(n) + n\log(\varepsilon + 1))$ expected time.*

In either case of (i) and (ii), the time complexity of ⑤ is bounded by $\mathcal{O}(n(1 + \log(\varepsilon + 1)))$. Therefore, as in Theorem 4.1, the overall time complexity of Index2Sort is proved to be $\mathcal{O}(nC(n) + nQ(n) + n\log(\varepsilon + 1))$. The detailed proof is provided in Appendix A.2.

| Index | $C(n),\ Q(n)$ | Assumption | Complexity of Index2Sort |
|---|---|---|---|
| - (Binary Search) | $C(n) = 0,\ Q(n) = \log n$ | - | $\mathcal{O}(n \log n)$ |
| B-tree | $C(n) = Q(n) = \log n$ | - | $\mathcal{O}(n \log n)$ |
| RDA Index | $C(n) = Q(n) = \log \log n$ | $\boldsymbol{D} \overset{\text{iid}}{\sim} \chi \in \mathfrak{X}_{\rho_1, \rho_2, \mathcal{K}}$ | $\mathcal{O}(n \log \log n)$ |
| ESPC Index | $C(n) = Q(n) = 1$ | $\boldsymbol{D} \overset{\text{iid}}{\sim} \chi \in \mathfrak{X}_{\rho_f, \mathcal{K}}$ | $\mathcal{O}(n)$ |
| ESPC Index | $C(n) = 1,\ Q(n) = \log \log n$ | $\boldsymbol{D} \overset{\text{iid}}{\sim} \chi \in \mathfrak{X}_C$ | $\mathcal{O}(n \log \log n)$ |
| Dynamic LI | $C(n) = Q(n) = \log \log n + \log(\delta n)$ | $\boldsymbol{D} \sim \boldsymbol{\chi} \subset \mathfrak{X}_{\rho_1, \rho_2, \mathcal{K}} \wedge \Delta(\boldsymbol{\chi}) \leq \delta$ | $\mathcal{O}(n \log \log n + n \log(\delta n))$ |

Table 1: Computational complexity of Index2Sort using various index structures: RDA Index (Zeighami & Shahabi, 2023), ESPC Index (Croquevielle et al., 2025), and Dynamic LI (Zeighami & Shahabi, 2024).

**Worst-Case Complexity.**  In addition to the expected time complexity analysis, we also provide the following theorem on the worst-case time complexity of Index2Sort.

> **Theorem 4.5.** *Consider a static index algorithm satisfying: (1) given a sorted array of length $n$, the index is constructed in $\mathcal{O}(nC(n))$ worst-case time. (2) the index answers a rank query in $\mathcal{O}(Q(n))$ worst-case time. Also, assume that the algorithm used for sorting each range bucket in the step ⑤ of Index2Sort has a worst-case time complexity of $\mathcal{O}(R(n))$ for sorting an array of length $n$, where $R(n)$ is a superadditive function, i.e., for any $n_1 \geq 0$ and $n_2 \geq 0$, $R(n_1 + n_2) \geq R(n_1) + R(n_2)$. Then, Index2Sort sorts an array of length $n$ in $\mathcal{O}(nC(n) + nQ(n) + R(n))$ worst-case time.*

This theorem implies that in a typical setting, where $C(n) = \log n$, $Q(n) = \log n$, and $R(n) = n \log n$, the worst-case time complexity of Index2Sort is $\mathcal{O}(n \log n)$. This matches the complexity of many classical comparison-based sorting algorithms. A rigorous proof is provided in Appendix A.3.

The $R(n)$ term is unavoidable in a worst-case bound. Indeed, all elements of $\boldsymbol{v}$ may fall into a single range bucket, in which case Index2Sort must explicitly sort a bucket of size $\Theta(n)$, costing $\mathcal{O}(R(n))$ time. The initial shuffle makes such an imbalance unlikely in the expected-time analysis, but it must still be accounted for in the worst case.

**Beyond Sorting.**  Furthermore, we show that, beyond sorting, the theoretical results for algorithms with predictions can be extended to a problem setting in which only the *algorithm for generating predictions* is given. The details of this generalization and the theoretical guarantees are provided in Appendix B.

## 4.3   Derived Computational Guarantees for Index2Sort

By the theorems introduced in Section 4.2, the computational guarantees of Index2Sort can be derived simply by substituting the cost functions $C(n)$ and $Q(n)$ of existing index data structures. Table 1 summarizes these derivations for several representative classical and learned indexes.

The primary advantage of this framework is its structure-agnostic nature. For instance, prior learned sorting algorithms (Sato & Matsui, 2025; Zeighami & Shahabi, 2024) require intricate analyses tailored to their specific algorithmic structures to achieve $\mathcal{O}(n \log \log n)$ complexity. In contrast, our framework obtains the same guarantee by treating the underlying index, namely the RDA Index, as a black box with known construction and query costs (Table 1, row 3).

As illustrative consequences, applying the ESPC-index (Croquevielle et al., 2025) yields $\mathcal{O}(n)$ complexity under the weaker assumption $\chi \in \mathfrak{X}_{\rho_f, \mathcal{K}}$, and $\mathcal{O}(n \log \log n)$ complexity under the even weaker assumption $\chi \in \mathfrak{X}_C$. The same substitution-based argument also extends to settings with distribution drift by using dynamic learned indexes (Zeighami & Shahabi, 2024). Detailed proofs and discussions for each instantiation are provided in Appendix D.

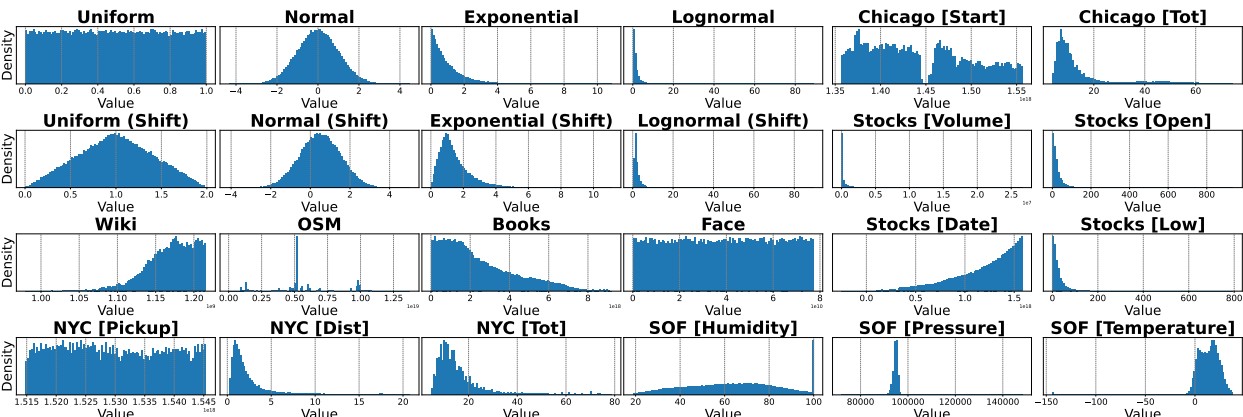

Figure 2: Histogram of the input values for each dataset.

## 5 Experiments

In this section, we experimentally validate the computational complexity derived in our theorems. We primarily focus on step ⑤ of Index2Sort, where the array $v$ is sorted using the index output (Sections 5.1 and 5.2). This focus is because, as noted in the proof sketch of Theorem 4.1, the complexities of other steps are obvious. The only non-trivial aspect is the complexity of step ⑤: it is $\mathcal{O}(n)$ with exact ranks (Theorems 4.1 to 4.3) and $\mathcal{O}(n(1 + \log(\varepsilon + 1)))$ with $\varepsilon$-approximate ranks (Theorem 4.4). We report the total number of comparisons accumulated across all executions of step ⑤, including those arising from recursive invocations triggered by step ②. Our primary metric is the number of comparisons, which is machine- and implementation-independent and directly corresponds to the theoretical complexity analyzed in our theorems. As a supplementary metric, we also report the sorting time to provide practical insight into runtime performance, although it depends on hardware and implementation details (Section 5.3).

All experiments were implemented in C++ and conducted on a single thread on a Linux machine equipped with an Intel® Core™ i9-11900H CPU @ 2.50 GHz and 64 GB of memory. The code was compiled using GCC version 9.4.0 with the `-O3` optimization flag. We report the mean and standard deviation over 10 runs for each data point in the figures.

**Datasets.** We used both artificial data and real-world data to support our theorems. Figure 2 shows the histogram of the input values for each dataset. For artificial data, we used the following four distributions and their *shifted* versions: **Uniform** distribution on $[0, 1]$, **Normal** distribution with $\mu = 0, \sigma = 1$, **Exponential** distribution with $\lambda = 1$, and **Lognormal** distribution with $\mu = 0, \sigma = 1$. The distribution shift was performed by adding $i/n$ to the $i$-th element.

For real-world data, we used the following 16 datasets commonly adopted in prior learned sorting studies: **Chicago [Start, Tot]** (Chicago, 2021), **Stocks [Volume, Open, Date, Low]** (Onyshchak, 2020), **Wiki**, **OSM**, **Books**, **Face** (Marcus et al., 2020a), **NYC [Pickup, Dist, Tot]** (nyc, 2020), and **SOF [Humidity, Pressure, Temperature]** (Mavrodiev, 2019). For each dataset, we construct the input array by randomly sampling $n$ keys and shuffling them.

**Choice of Index Structures.** We select index structures to validate the primary claim of Index2Sort: that the computational guarantees of a static index transfer to sorting. We therefore include representative and widely used indexes, covering both classical baselines and indexes with explicit construction and query complexity guarantees: (i) **B-tree**, the classical static index with worst-case guarantees, (ii) **RMI**, the first learned index and a standard baseline (Kraska et al., 2018), (iii) **PGM-index**, a learned index with strong worst-case complexity guarantees (Ferragina & Vinciguerra, 2020), and (iv) **ESPC index**, a learned index with state-of-the-art expected-time guarantees (Croquevielle et al., 2025). We also include a non-index

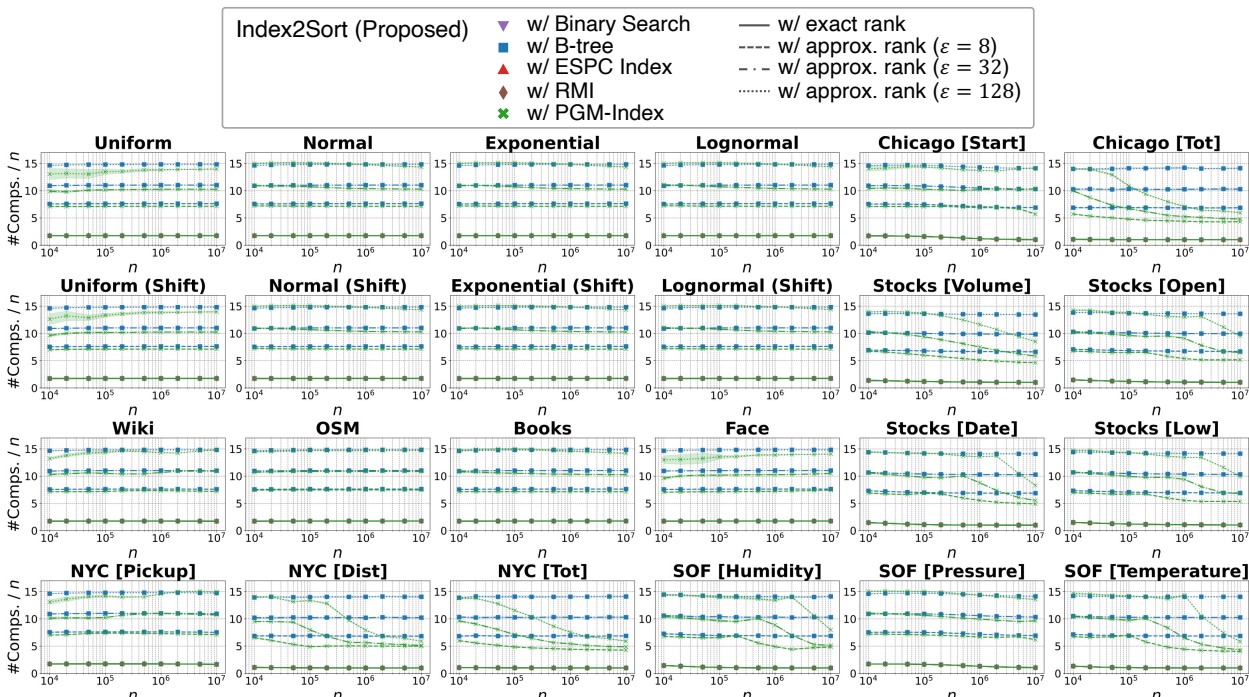

Figure 3: Total number of element comparisons performed in step ⑤ of Index2Sort (including all recursive calls) for input length $n$. Across distributions, index types, and rank precision (exact or approximate), the total number of comparisons scales linearly with $n$.

baseline that answers rank queries via **binary search**; in this case, Index2Sort is essentially equivalent to classical Index Sort (Gurram & Gera, 2011).

## 5.1 Scaling of Comparisons in Step ⑤ with $n$

We experimentally show that the number of comparisons in ⑤ of Index2Sort grows linearly with the input array length $n$ under various conditions. In this experiment, we set $\tau = 128$ and $\alpha = 1/2$, and approximate ranks were obtained using exponential search. Figure 3 plots the number of comparisons in ⑤ against $n$. The results show that, in most cases, the number of comparisons grows almost perfectly linearly with $n$. When using a B-tree as the underlying index, this linear trend is nearly exact. In contrast, with a PGM-index, the number of comparisons sometimes grows at a rate slightly slower than linear. This effect is particularly noticeable for datasets with many duplicate values, where the PGM-index resolves duplicates more efficiently than a B-tree and answers approximate rank queries with fewer errors, thereby reducing the cost of exponential search in step ⑤. Overall, the number of comparisons exhibits linear growth in $n$, regardless of the distribution (including shifted and real-world data), the index type, or the rank precision (exact or approximate), supporting Theorems 4.1 to 4.4.

## 5.2 Scaling of Comparisons in Step ⑤ with $\varepsilon$

We also show that the number of comparisons in ⑤ scales proportionally to $\log \varepsilon$. We again set $\tau = 128$ and $\alpha = 1/2$, and fix the array length to $n = 10^7$. As before, approximate ranks were computed via exponential search. Figure 4 shows the number of comparisons in ⑤ when using a B-tree or a PGM-index with a maximum error of $\varepsilon$. We observe that, in most cases, the number of comparisons scales proportionally to $\log \varepsilon$. For the B-tree, this proportionality is nearly exact. With a PGM-index, especially on datasets with many duplicate values, the number of comparisons can be noticeably smaller than in the B-tree case, since improved rank accuracy reduces the cost of exponential search in step ⑤. These results confirm proportionality to $\log \varepsilon$, regardless of the distribution, distribution shifts, or index type, supporting Theorem 4.4.

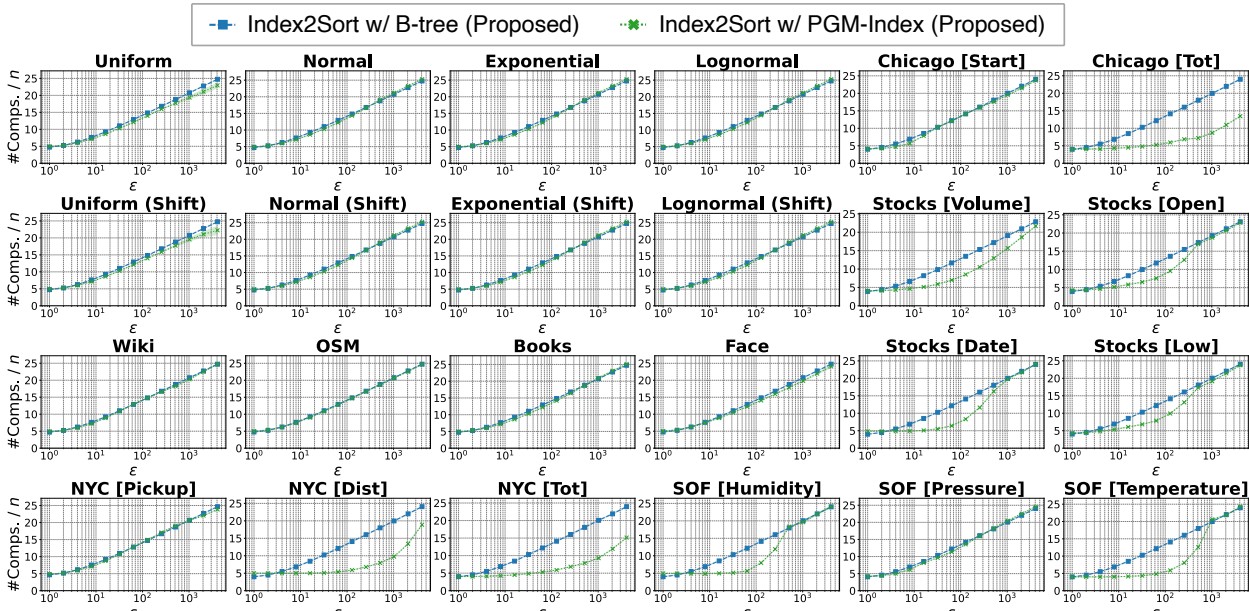

Figure 4: Total number of element comparisons performed in step ⑤ of Index2Sort (including all recursive calls) when using an index with maximum rank error $\varepsilon$. Across distributions and index types, the total number of comparisons is proportional to $\log \varepsilon$.

## 5.3 Sorting Time

Here, we compare the actual sorting time of Index2Sort against a broad set of baselines. In the main text, we report `std::sort` (IntroSort) (Musser, 1997) and IS[4]o (Axtmann et al., 2022) as classical baselines, and Learned Sort 2.0 (Kristo et al., 2021) and Learned Sort 2.1 (Ferragina & Odorisio, 2025) as learned sorting baselines. For Index2Sort, we report the results of the variants built on the ESPC index and the RMI. Additional baselines and index choices are evaluated in Appendix F. We set $\alpha = 1/32$ and use approximate ranks ($\varepsilon = 64$). For step ⑤, we use a hybrid post-processing scheme to improve practical performance without affecting the theoretical guarantees: we bucket elements by predicted ranks and sort each bucket with `std::sort`, then run an insertion-style pass capped at $\Theta(n)$ swaps; if the array remains unsorted, we invoke the modified Displacement Sort.

Figure 5 plots the sorting time versus input length $n$. Overall, Index2Sort with the ESPC index is the fastest among our Index2Sort variants, and its growth is much slower than $n \log n$ (e.g., `std::sort`), indicating an expected complexity of $o(n \log n)$. On artificial data, the results match our theory: for inputs drawn from the uniform distribution (which falls into $\mathfrak{X}_{\rho_f, \mathcal{K}}$), the running time is $\mathcal{O}(n)$, whereas for inputs drawn from the normal or exponential distributions (which fall into $\mathfrak{X}_C$), it is $\mathcal{O}(n \log \log n)$. Although the log-normal distribution is not covered by $\mathfrak{X}_C$, Index2Sort with the ESPC index still exhibits a gradual growth trend in practice. We further verified that Index Sort runs in expected $\mathcal{O}(n \log n)$ time regardless of the distribution, and that Index2Sort with either a PGM-index or a B-tree has expected $\mathcal{O}(n \log n)$ complexity.

We also observe that Index2Sort inherits performance characteristics of the underlying index. For example, it is known that the RMI can suffer a slowdown on the OSM dataset (Marcus et al., 2020b), and we found the same trend for Index2Sort built on the RMI, which also slows down relatively on OSM.

Our ESPC-based Index2Sort is on average slower than IS[4]o and recent learned sorters (Learned Sort 2.1), which is expected: those systems are heavily tuned for cache efficiency and other low-level effects, whereas our implementation prioritizes rigorous guarantees. Quantitatively, ESPC-based Index2Sort is slower than the fastest sorter by roughly 2.0× on average and by at most 4.2× in our experiments. A cache-aware, highly optimized implementation of Index2Sort is left for future work.

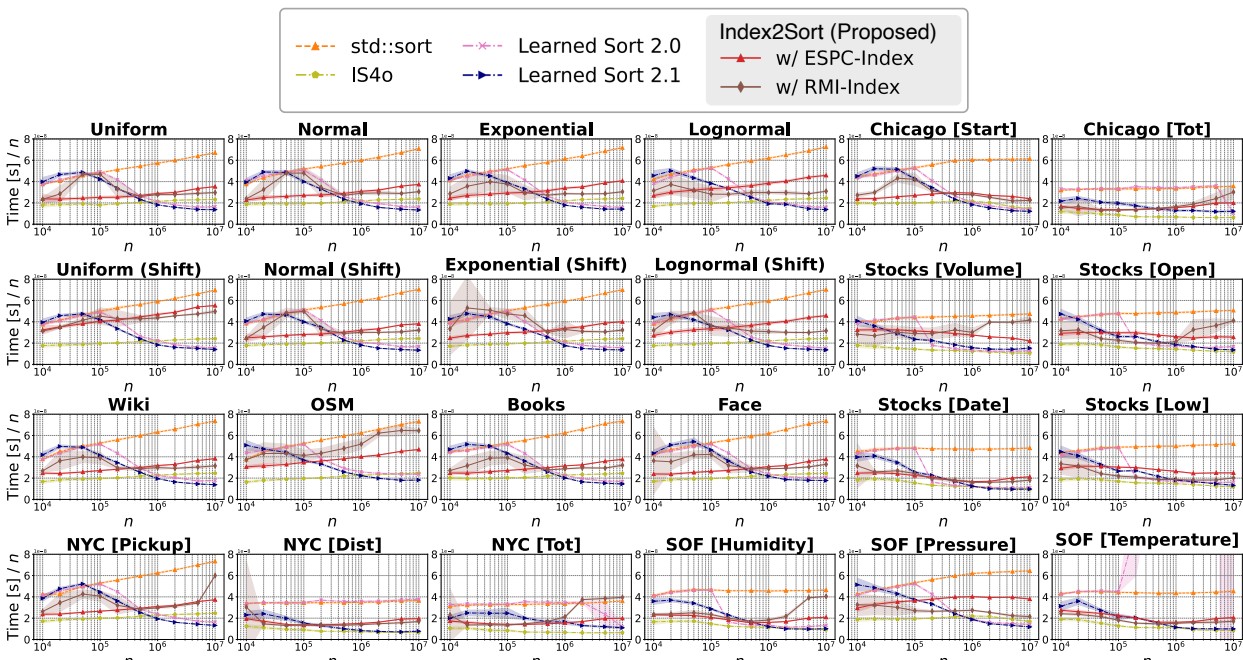

Figure 5: Sorting time vs. input length $n$. Index2Sort with the ESPC index scales substantially slower than $n \log n$ in practice, whereas learned sorting algorithms without strong complexity guarantees exhibit severe slowdowns on some datasets.

Finally, the experiments highlight the risk of weak worst-case guarantees. For example, on SOF [Temperature] with $n = 10^5$, ESPC-based Index2Sort finished within 0.023 seconds, while Learned Sort 2.0, which has $\mathcal{O}(n^2)$ worst-case time, took up to 108.9 seconds. This demonstrates the importance of having strong worst-case guarantees for learned sorting algorithms.

## 6 Limitations and Future Work

While our analysis provides strong guarantees in terms of expected running time, the worst-case complexity of Index2Sort is $\mathcal{O}(nC(n) + nQ(n) + R(n))$, where $R(n)$ is the worst-case complexity of the sorting algorithm applied to each range bucket. In practice, this still provides strong protection against slowdowns: under typical settings, it simplifies to $\mathcal{O}(n \log n)$, matching the bounds for classical comparison-based sorting and preventing catastrophic performance degradation. Nonetheless, the explicit dependence on $R(n)$ points to a natural direction for future work: can this dependence be removed through a more refined analysis or alternative algorithmic designs? Eliminating it could yield tighter worst-case guarantees and further strengthen the theoretical foundation of Index2Sort.

Another interesting open problem is developing a *Sort2Index* framework. Specifically, given a sorting algorithm, can we construct an indexing algorithm based on that sorting algorithm and provide theoretical guarantees on its computational complexity? Although the equivalence between sorting and priority queues has been established (Thorup, 2007), two key differences between priority queues and static indexes make this problem worth investigating: (1) priority queues are dynamic, while static indexes are static, and (2) priority queues support only minimum value extraction operations, while static indexes answer rank queries.

Finally, we note that our contributions are primarily theoretical. Although Index2Sort provides stronger asymptotic guarantees than existing sorting algorithms, it does not necessarily outperform them in practical runtime due to the lack of low-level hardware optimizations. Bridging this theory-practice gap through hardware-conscious design or implementation-level optimizations is an important direction for future work.

# 7    Conclusion

In this paper, we proposed Index2Sort, a general framework for deriving sorting algorithms from static indexes. We proved that Index2Sort automatically inherits the computational guarantees of the underlying index, yielding strictly stronger complexity bounds than existing learned sorts. This work bridges the gap between theory on learned indexes and learned sorts, enabling future advances in index research to be transferred directly to sorting.

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

# A    Proofs

Here, we provide the proofs omitted in the main text. First, in Appendix A.1, we present the proofs of the three fundamental theorems of Index2Sort: Theorems 4.1 to 4.3. Next, in Appendix A.2, we detail the necessary modifications to the Index2Sort algorithm for handling approximate rank queries and prove the corresponding time complexity guarantee stated in Theorem 4.4. Finally, in Appendix A.3, we introduce and prove Theorem 4.5, which establishes the worst-case time complexity of Index2Sort. While the main text explains the algorithm assuming $\alpha = 1/2$ for simplicity, in the following, we generalize the analysis to allow the number of buckets, $m$, to be defined as $m = \lfloor \alpha n \rfloor$ for any constant $\alpha \in (0,1)$. Additionally, while the main text describes MergeSort as the algorithm used for sorting range buckets, in the following, we allow any sorting algorithm with a time complexity of $\mathcal{O}(n^2)$. All expectations and probabilities in our analysis are computed with respect to the random shuffle performed by the Index2Sort algorithm (①). Additionally, if the input array or query is assumed to be drawn from a certain distribution, the corresponding sampling randomness is also included in our analysis.

## A.1    Proof of Theorems 4.1 to 4.3

Here, we first present Theorem A.1, a lemma that provides the theoretical guarantees for the bucket sorting step in Index2Sort. We then use this lemma to prove the theorems Theorems 4.1 to 4.3.

**Lemma A.1.** *The expected time complexity for the step ⑤ of Index2Sort is $\mathcal{O}(n)$.*

*Proof of Theorem A.1.* Let $A_i$ ($i \in \{1, 2, \ldots, m+1\}$) denote the number of elements in the $i$-th range bucket when Index2Sort is applied to an input array of length $n$. Specifically, $A_i$ denotes the number of elements in the $i$-th range bucket $\boldsymbol{r}_i$, obtained by bucketing the array $\boldsymbol{v} \in \mathbb{R}^{n-m}$ using the thresholds $\boldsymbol{u} \in \mathbb{R}^m$ while using the point-bucket mechanism.

Additionally, let $B_i$ ($i \in \{1, 2, \ldots, m+1\}$) denote the number of elements in the $i$-th bucket obtained by bucketing the same $\boldsymbol{v} \in \mathbb{R}^{n-m}$ using the same thresholds $\boldsymbol{u} \in \mathbb{R}^m$ **without** applying the point bucket mechanism; that is, each element is always assigned to exactly one of the $m+1$ buckets. Here, to disambiguate the handling of values equal to any threshold in $\boldsymbol{u}$, we associate each input element $x_i$ with its original index $i$, forming tuples $(x_i, i)$. These tuples are then totally ordered, so ties in value are resolved by input order. Bucketing is performed straightforwardly according to this total order, allowing us to utilize the results of (Frazer & McKellar, 1970) for analysis of $\boldsymbol{B}$.

Then, for the same $\boldsymbol{u}$ and $\boldsymbol{v}$, $A_i \le B_i$ holds for all $i$. This is because if $v_j$ ($j \in \{1, 2, \ldots, n-m\}$) falls into the $i$-th range bucket $\boldsymbol{r}_i$ using the Index2Sort method, then $v_j$ will also fall into the $i$-th bucket in the bucketing procedure described in the definition of $\boldsymbol{B}$. Therefore,

$$\mathbb{E}\left[\sum_{i=1}^{m+1} A_i{}^2\right] = \sum_{i=1}^{m+1} \mathbb{E}\left[A_i{}^2\right] \tag{1}$$

$$\le \sum_{i=1}^{m+1} \mathbb{E}\left[B_i{}^2\right]. \tag{2}$$

Now, from Lemma 1 in (Frazer & McKellar, 1970), $\Pr[B_i = j] = \binom{n-j-1}{m-1}/\binom{n}{m}$. Therefore,

$$\sum_{i=1}^{m+1} E\left[B_i{}^2\right] = \sum_{i=1}^{m+1}\sum_{j=0}^{n-m} j^2 \Pr\left[B_i = j\right] \tag{3}$$

$$= \sum_{i=1}^{m+1}\sum_{j=0}^{n-m} j^2 \cdot \frac{\binom{n-j-1}{m-1}}{\binom{n}{m}} \tag{4}$$

$$= \frac{m+1}{\binom{n}{m}}\sum_{j=0}^{n-m} j^2 \binom{n-j-1}{m-1}. \tag{5}$$

Here, we evaluate the sum as follows:

$$\sum_{j=0}^{n-m} j^2 \binom{n-j-1}{m-1} \tag{6}$$

$$= \sum_{k=m-1}^{n-1} (n-1-k)^2 \binom{k}{m-1} \qquad (k := n-j-1) \tag{7}$$

$$= (n-1)^2 \sum_{k=m-1}^{n-1} \binom{k}{m-1} - 2(n-1) \sum_{k=m-1}^{n-1} k \binom{k}{m-1} + \sum_{k=m-1}^{n-1} k^2 \binom{k}{m-1} \tag{8}$$

$$= (n-1)^2 \binom{n}{m} - 2(n-1)\left( (m-1)\binom{n}{m} + m\binom{n}{m+1} \right) + \left( (m-1)^2 \binom{n}{m} \right. \tag{9}$$

$$\left. + m(2m-1)\binom{n}{m+1} + m(m+1)\binom{n}{m+2} \right) \qquad (\because \text{Hockey-stick identity}) \tag{10}$$

$$= (n-m)^2 \binom{n}{m} + m(2m-2n+1)\binom{n}{m+1} + m(m+1)\binom{n}{m+2} \tag{11}$$

$$= \left( (n-m)^2 + m(2m-2n+1) \cdot \frac{n-m}{m+1} + m(m+1) \cdot \frac{(n-m)(n-m-1)}{(m+2)(m+1)} \right) \binom{n}{m} \tag{12}$$

$$= \frac{(n-m)(2n-m)}{(m+1)(m+2)} \binom{n}{m}. \tag{13}$$

Therefore,

$$\sum_{i=1}^{m+1} E\left[ B_i{}^2 \right] = \frac{m+1}{\binom{n}{m}} \cdot \frac{(n-m)(2n-m)}{(m+1)(m+2)} \binom{n}{m} \tag{14}$$

$$= \frac{(n-m)(2n-m)}{m+2}. \tag{15}$$

With $m = \lfloor \alpha n \rfloor$ for a fixed $\alpha \in (0,1)$, this expression is $\Theta(n)$. Therefore,

$$\mathbb{E}\left[ \sum_{i=1}^{m+1} A_i{}^2 \right] = \mathcal{O}(n). \tag{16}$$

Therefore, since Index2Sort uses a sorting algorithm with $\mathcal{O}(n^2)$ time complexity for sorting each range bucket, the expected time complexity for the step ⑤ is $\mathcal{O}(n)$. $\qquad\square$

Using this lemma, we provide the proofs for Theorems 4.1 to 4.3.

*Proof of Theorem 4.1.* Let the expected time complexity of Index2Sort be $S(n)$. By mathematical induction, we show that $S(n) = \mathcal{O}(nC(n) + nQ(n) + n)$. The time complexity of each step in the Index2Sort algorithm is as follows:

① Splitting the array requires $\mathcal{O}(n)$ computations. The initial shuffle is also $\mathcal{O}(n)$.

② Recursively sorting $\boldsymbol{u}$ requires $S(\alpha n)$.

③ Constructing the index on $\boldsymbol{u}$ has an expected complexity of $\mathcal{O}(\alpha n C(\alpha n))$.

④ Answering rank queries for all elements of $\boldsymbol{v}$ using the index requires $\mathcal{O}((1-\alpha)nQ(\alpha n))$ expected complexity.

⑤ Sorting $\boldsymbol{v}$ using the results of rank queries has an expected complexity of $\mathcal{O}(n)$ (by Theorem A.1).

⑥ Merging $\boldsymbol{u}'$ and $\boldsymbol{v}'$ requires $\mathcal{O}(n)$.

For ⑤, we rely on Theorem A.1 and the assumption that the sorting algorithm used for range buckets runs in $\mathcal{O}(n^2)$. Thus, $S(n)$ can be expressed recursively as follows:

$$S(n) = S(\alpha n) + \mathcal{O}(\alpha n C(\alpha n) + (1 - \alpha)n Q(\alpha n) + n). \tag{17}$$

Here, suppose there exist constants $c \in \mathbb{R}_{>0}$ and $n_0 \in \mathbb{N}$ such that for any $n_0 \le n' < n$, we have:

$$S(n') \le c(n' C(\alpha n') + n' Q(\alpha n') + n'). \tag{18}$$

In the following, we show that by taking a sufficiently large constant $c$ (which does not depend on $n$), we can obtain $S(n) \le c(n C(\alpha n) + n Q(\alpha n) + n)$. We consider two cases: when $\alpha n \ge n_0$ and when $\alpha n < n_0$.

In the first case, i.e., when $\alpha n \ge n_0$, from Equation (18), it follows that

$$S(\alpha n) \le c(\alpha n C(\alpha^2 n) + \alpha n Q(\alpha^2 n) + \alpha n). \tag{19}$$

Therefore, from Equation (17), we get

$$S(n) \le c(\alpha n C(\alpha^2 n) + \alpha n Q(\alpha^2 n) + \alpha n) + \mathcal{O}(\alpha n C(\alpha n) + (1 - \alpha)n Q(\alpha n) + n). \tag{20}$$

Here, by defining $\beta := \max(\alpha, 1 - \alpha)$ and taking $c$ sufficiently large, we can rewrite this as:

$$S(n) \tag{21}$$
$$\le c(\beta n C(\alpha^2 n) + \beta n Q(\alpha^2 n) + \beta n) + \mathcal{O}(\beta n C(\alpha n) + \beta n Q(\alpha n) + n) \tag{22}$$
$$\le c(\beta n C(\alpha n) + \beta n Q(\alpha n) + \beta n) + \mathcal{O}(n C(\alpha n) + n Q(\alpha n) + n) \tag{23}$$
$$= c(n C(\alpha n) + n Q(\alpha n) + n) - c(1 - \beta)(n C(\alpha n) + n Q(\alpha n) + n) + \mathcal{O}(n C(\alpha n) + n Q(\alpha n) + n) \tag{24}$$
$$\le c(n C(\alpha n) + n Q(\alpha n) + n). \tag{25}$$

In the second inequality, we use the fact that $C$ and $Q$ are non-decreasing functions of $n$. The final inequality holds by choosing $c$ as a sufficiently large constant.

In the second case, i.e., $\alpha n < n_0$, there exists a certain constant $d > 0$, which does not depend on $n$, such that

$$S(\alpha n) \le d. \tag{26}$$

This is because, since $\alpha n < n_0$, $S(\alpha n)$ is at most $\max_{n' \in \{1, \dots, n_0 - 1\}} S(n')$, which does not depend on $n$. Therefore, from Equation (17),

$$S(n) \le d + \mathcal{O}(\alpha n C(\alpha n) + (1 - \alpha)n Q(\alpha n) + n). \tag{27}$$

Since $n C(\alpha n) + n Q(\alpha n) + n \ge 1$, by taking $c$ sufficiently large,

$$d + \mathcal{O}(\alpha n C(\alpha n) + (1 - \alpha)n Q(\alpha n) + n) \le c(n C(\alpha n) + n Q(\alpha n) + n). \tag{28}$$

Therefore, from Equations (27) and (28), we get $S(n) \le c(n C(\alpha n) + n Q(\alpha n) + n)$.

By mathematical induction, we conclude that for any $n \ge n_0$, $S(n) \le c(n C(\alpha n) + n Q(\alpha n) + n)$. Thus, we have $S(n) = \mathcal{O}(n C(\alpha n) + n Q(\alpha n) + n)$. Since $C$ and $Q$ are non-decreasing functions and $Q(n) \ge 1$, we deduce that $S(n) = \mathcal{O}(n C(n) + n Q(n))$. $\qquad\square$

*Proof of Theorem 4.2.* Under the assumption that each element of the input array is sampled i.i.d. from a single distribution $\chi \in \mathfrak{X}$, let the expected time complexity of Index2Sort be $S(n)$. Following an approach similar to the proof of Theorem 4.1, the time complexity of each step in the Index2Sort algorithm is as follows:

① As in Theorem 4.1, the complexity is $\mathcal{O}(n)$.

② Sorting $\boldsymbol{u}$ recursively takes $S(\alpha n)$, since the elements of $\boldsymbol{u}$ are sampled i.i.d. from the same distribution $\chi$.

③ Constructing the index on $\boldsymbol{u}$ runs in $\mathcal{O}(\alpha n C(\alpha n))$, since $\boldsymbol{u}'$ is a sorted version of $\boldsymbol{u}$, an array sampled i.i.d. from the distribution $\chi$.

④ Answering rank queries for all elements of $\boldsymbol{v}$ using the index requires $\mathcal{O}((1-\alpha)nQ(\alpha n))$, since $\boldsymbol{u}'$ is a sorted version of $\boldsymbol{u}$, an array sampled i.i.d. from the distribution $\chi$, and $\boldsymbol{v}$ is also independently sampled from $\chi$.

⑤ As in Theorem 4.1, the complexity is $\mathcal{O}(n)$.

⑥ As in Theorem 4.1, the complexity is $\mathcal{O}(n)$.

For the steps ②, ③, and ④, the propagation of the i.i.d. assumption from $\boldsymbol{x}$ to $\boldsymbol{u}$ and $\boldsymbol{v}$ is utilized. Specifically, the assumption that each element of $\boldsymbol{x}$ is sampled i.i.d. from $\chi \in \mathfrak{X}$ ensures that each element of $\boldsymbol{u}$ and $\boldsymbol{v}$ is also sampled i.i.d. from $\chi \in \mathfrak{X}$. This satisfies the assumptions required for the time complexity guarantees of both Index2Sort and the index, allowing the respective guarantees to be applied. For the step ⑤, since Theorem A.1 does not rely on any distributional assumptions, the complexity remains the same as in Theorem 4.1. Therefore, $S(n)$ can be expressed using the same recurrence relation as in Theorem 4.1 (Equation (17)), leading to the same result, $S(n) = \mathcal{O}(nC(n) + nQ(n))$. □

*Proof of Theorem 4.3.* Under the assumption that each element of the input array is independently sampled from a sequence of distributions $\boldsymbol{\chi}$, where $\boldsymbol{\chi}$ has at most $\delta$ distribution shift and $\boldsymbol{\chi} \subset \mathfrak{X}$, let the expected time complexity of Index2Sort be $S(n)$. Following an approach similar to the proof of Theorem 4.1, the time complexity of each step in the Index2Sort algorithm is as follows:

① As in Theorem 4.1, the complexity is $\mathcal{O}(n)$.

② Sorting $\boldsymbol{u}$ recursively takes $S(\alpha n)$ since the elements of $\boldsymbol{u}$ are independently sampled from a sequence of distributions with at most $\delta$ distribution shift.

③ Constructing the index on $\boldsymbol{u}$ runs in $\mathcal{O}(\alpha n C(\alpha n, \delta))$, since $\boldsymbol{u}'$ is a sorted version of $\boldsymbol{u}$, which is sampled independently from a sequence of distributions with at most $\delta$ distribution shift.

④ Answering rank queries for all elements of $\boldsymbol{v}$ using the index requires $\mathcal{O}((1-\alpha)nQ(\alpha n, \delta))$, because $\boldsymbol{u}'$ is a sorted version of $\boldsymbol{u}$, which is sampled independently from a sequence of distributions with at most $\delta$ distribution shift, and $\boldsymbol{v}$ is also sampled independently from distributions in $\boldsymbol{\chi}$ with at most $\delta$ distribution shift.

⑤ As in Theorem 4.1, the complexity is $\mathcal{O}(n)$.

⑥ As in Theorem 4.1, the complexity is $\mathcal{O}(n)$.

For the steps ②, ③, and ④, the assumption that the elements of $\boldsymbol{x}$ are independently sampled from a sequence of distributions $\boldsymbol{\chi}$ with at most $\delta$ distribution shift ensures that the elements of $\boldsymbol{u}$ and $\boldsymbol{v}$ also follow the same assumption. This allows the time complexity guarantees of both Index2Sort and the index to be applied recursively. For the step ⑤, since Theorem A.1 does not rely on any distributional assumptions, the complexity remains the same as in Theorem 4.1. Thus, $S(n)$ can be expressed as follows:

$$S(n) = S(\alpha n) + \mathcal{O}(\alpha n C(\alpha n, \delta) + (1-\alpha)nQ(\alpha n, \delta) + n), \tag{29}$$

leading to the result, $S(n) = \mathcal{O}(nC(n, \delta) + nQ(n, \delta))$. □

### A.2 Proof of Theorem 4.4

Next, we prove Theorem 4.4, which demonstrates that Index2Sort remains valid even under the condition that approximate rank queries are allowed. For the guarantees on time complexity, it is necessary to implement one of the two algorithmic modifications mentioned in the main text; (i) Instead of bucket sorting, use a slightly modified version of the sorting with predictions algorithm (Bai & Coester, 2023) to sort $\boldsymbol{v}$, or (ii) Perform an exponential search on $\boldsymbol{u}'$ to determine the exact rank using the approximate rank query result as the starting point. Here, we first present two key lemmas, Theorem A.2 and Theorem A.3, which are critical for guaranteeing the time complexity when modification of (i) is applied. We then show that regardless of whether modification (i) or (ii) is applied, the time complexity of Index2Sort is bounded as stated in Theorem 4.4.

**Modified Displacement Sort**  Displacement Sort, the sorting with predictions algorithm proposed in (Bai & Coester, 2023), is a simple yet effective approach that proceeds as follows:

1. Assign each element to the bucket according to the prediction, which is the predicted position in the sorted array.

2. Insert elements from buckets with smaller predicted values sequentially into a data structure called a finger tree (Guibas et al., 1977).

3. Extract values from the finger tree in increasing order to obtain the sorted array.

A finger tree is a binary tree with a "finger," a pointer to the most recently accessed or inserted element. This structure enables fast access and insertion of elements near the finger. Specifically, accessing or inserting an element at a distance $d$ from the finger can be done in $\mathcal{O}(\log d)$ time. In (Bai & Coester, 2023), this property is leveraged to achieve very low time complexity when the predictions are reasonably accurate.

To make this algorithm applicable to Index2Sort, we introduce the following two minor modifications:

- Extension of the prediction range: In the original algorithm, the prediction range was defined as $\{1, 2, \ldots, l\}$ for an input array of length $l$. We extend this range to a contiguous set of $\Theta(l)$ integers (in our case, the prediction is the approximate rank, so it is in $\{0, 1, 2, \ldots, m\}$). Concretely, we prepare buckets corresponding to each of these $\Theta(l)$ integers and assign elements to buckets based on their predicted values.

- Modification for duplicate handling: Instead of storing only the values in each node of the finger tree, we modify the structure to store both the value and its frequency (i.e., the number of times it has appeared). When inserting a value into the finger tree, if the value already exists, we simply increment its frequency instead of adding a new node.

The first modification is necessary in Index2Sort because the size of the prediction range, $m + 1$, does not necessarily match the length of the array to be sorted, $n - m$. The second modification is required because the original Displacement Sort algorithm assumes there are no duplicate elements in the input array, whereas Index2Sort considers the possibility of duplicate elements.

Now, we provide a theoretical guarantee for the extended Displacement Sort algorithm described above. Consider an input array $\boldsymbol{v} \in \mathbb{R}^l$ of length $l$ with predictions $\hat{\boldsymbol{p}} \in \{1, 2, \ldots, m+1\}^l$ (where $m = \Theta(l)$). Define the prediction error metric $\eta_i \in \mathbb{N}$ for each element $x_i$ as follows: $\eta_i = |\{v_j \mid j \in \{1, 2, \ldots, l\} \land v_i \leq v_j \land \hat{p}_j \leq \hat{p}_i\}|$. Then, the following lemma holds:

**Lemma A.2.** *The time complexity of the Displacement Sort algorithm, extended as described above, for sorting the array $\boldsymbol{v}$ is $\mathcal{O}(l + \sum_{i=1}^{l} \log(\eta_i + 1))$.*

*Proof of Theorem A.2.* First, the computational cost of distributing $\boldsymbol{v}$ into buckets using predicted values is $\mathcal{O}(l)$. This is because the number of buckets is $m + 1 = \Theta(l)$, and assigning each element to a bucket takes $\mathcal{O}(1)$.

Next, consider the time complexity of inserting elements into the finger tree. Let the concatenated array of the distributed buckets be $\boldsymbol{w} \in \mathbb{R}^l$. Define $d_i$ ($i \in \{2, 3, \ldots, l\}$) as the number of unique elements in $\{w_1, \ldots, w_{i-1}\}$ that fall within the closed interval of $[w_{i-1}, w_i]$ or $[w_i, w_{i-1}]$, i.e.,

$$d_i = |\{w_j \mid j \in \{1, 2, \ldots, i-1\} \wedge (w_j \in [w_{i-1}, w_i] \vee w_j \in [w_i, w_{i-1}])\}|. \tag{30}$$

When inserting $w_i$ ($i \in \{2, 3, \ldots, l\}$) into the finger tree, the computational cost is $\mathcal{O}(\log d_i)$. This follows from the properties of the finger tree and the fact that only unique elements are inserted into the finger tree, thanks to our extensions. Thus, the total computational cost is $\mathcal{O}(\sum_{i=2}^{l} \log d_i)$. The value of $d_i$ is bounded as follows:

$$d_i = |\{w_j \mid j \in \{1, 2, \ldots, i-1\} \wedge (w_j \in [w_{i-1}, w_i] \vee w_j \in [w_i, w_{i-1}])\}| \tag{31}$$
$$\leq |\{w_j \mid j \in \{1, 2, \ldots, i-1\} \wedge (w_{i-1} \leq w_j \vee w_i \leq w_j)\}| \tag{32}$$
$$\leq |\{w_j \mid j \in \{1, 2, \ldots, i-1\} \wedge w_{i-1} \leq w_j\}| + |\{w_j \mid j \in \{1, 2, \ldots, i-1\} \wedge w_i \leq w_j\}| \tag{33}$$
$$\leq |\{w_j \mid j \in \{1, 2, \ldots, i-2\} \wedge w_{i-1} \leq w_j\}| + |\{w_j \mid j \in \{1, 2, \ldots, i-1\} \wedge w_i \leq w_j\}| + 1. \tag{34}$$

Now, from the definition of $\eta_i$,

$$|\{w_j \mid j \in \{1, 2, \ldots, i-1\} \wedge w_i \leq w_j\}| \leq |\{v_j \mid j \in \{1, 2, \ldots, l\} \wedge v_i \leq v_j \wedge \hat{p}_j \leq \hat{p}_i\}| \tag{35}$$
$$= \eta_i. \tag{36}$$

Therefore, we have

$$d_i \leq \eta_{i-1} + \eta_i + 1. \tag{37}$$

The total computational cost of inserting elements into the finger tree is then:

$$\sum_{i=2}^{l} \mathcal{O}(\log d_i) \leq \sum_{i=2}^{l} \mathcal{O}(\log(\eta_{i-1} + \eta_i + 1)) \tag{38}$$

$$\leq \sum_{i=1}^{l} \mathcal{O}(\log(\eta_i + 1)). \tag{39}$$

Finally, extracting elements from the finger tree in sorted order takes at most $\mathcal{O}(l)$. Thus, the total time complexity of the modified Displacement Sort algorithm for sorting $\boldsymbol{v}$ is $\mathcal{O}(l + \sum_{i=1}^{l} \log(\eta_i + 1))$. $\qquad \square$

**Displacement Sort Complexity in Index2Sort**  Next, to analyze the time complexity of sorting $\boldsymbol{v}$ in Index2Sort using the modified Displacement Sort described above, we present the following lemma.

**Lemma A.3.** *In Index2Sort, when the results of approximate rank queries on $\boldsymbol{u}'$ (with at most $\varepsilon$ error) are used as predictions, the expected time complexity of sorting $\boldsymbol{v}$ using the modified Displacement Sort is $\mathcal{O}(n + n \log(\varepsilon + 1))$.*

*Proof of Theorem A.3.* In Index2Sort, the length of the array $\boldsymbol{v}$ to be sorted by the modified Displacement Sort is $n - m$. Let $l = n - m$.

Let $\hat{\boldsymbol{p}} \in \{1, 2, \ldots, m+1\}^l$ be the vector of approximate rank query results on $\boldsymbol{u}'$, and let $\boldsymbol{p} \in \{1, 2, \ldots, m+1\}^l$ be the vector of exact rank query results. Since the approximate rank query has at most $\varepsilon$ error, we have $|\hat{p}_i - p_i| \leq \varepsilon$ for any $i \in \{1, 2, \ldots, m+1\}$.

Let $\boldsymbol{\eta}$ be defined as in Theorem A.2, where $\eta_i = |\{v_j \mid j \in \{1, 2, \ldots, l\} \wedge v_i \leq v_j \wedge \hat{p}_j \leq \hat{p}_i\}|$. The time complexity of the modified Displacement Sort is $\mathcal{O}(l + \sum_{i=1}^{l} \log(\eta_i + 1))$.

Next, we bound $\eta_i$ as follows:

$$\eta_i = |\{v_j \mid j \in \{1, 2, \ldots, l\} \wedge v_i \leq v_j \wedge \hat{p}_j \leq \hat{p}_i\}| \tag{40}$$

$$\leq |\{v_j \mid j \in \{1, 2, \ldots, l\} \wedge v_i \leq v_j \wedge p_j \leq p_i + 2\varepsilon\}| \tag{41}$$

$$\leq |\{v_j \mid j \in \{1, 2, \ldots, l\} \wedge p_i \leq p_j \leq p_i + 2\varepsilon\}| \tag{42}$$

$$\leq \sum_{r=p_i}^{p_i+2\varepsilon} |\{v_j \mid j \in \{1, 2, \ldots, l\} \wedge p_j = r\}|. \tag{43}$$

The first inequality uses the fact that $\hat{p}_j$ differs from $p_j$ by at most $\varepsilon$, while the second uses $v_i \leq v_j \Rightarrow p_i \leq p_j$.

Now, consider $|\{v_j \mid j \in \{1, 2, \ldots, l\} \wedge p_j = r\}|$, i.e., the number of unique elements in $\boldsymbol{v}$ whose exact rank in $\boldsymbol{u}'$ is $r$. Let the indices of elements selected as $\boldsymbol{u}$ in the sorted array $\boldsymbol{x}'$ be $i_1, i_2, \ldots, i_m$, where $1 \leq i_1 < i_2 < \cdots < i_m \leq n$. Additionally, let $i_0 = 0$, $i_{m+1} = n + 1$, $x'_0 = -\infty$, and $x'_{n+1} = \infty$. Then, the value $|\{v_j \mid j \in \{1, 2, \ldots, l\} \wedge p_j = r\}|$ can be bounded as follows:

$$|\{v_j \mid j \in \{1, 2, \ldots, l\} \wedge p_j = r\}| \leq |\{x_j \mid j \in \{1, 2, \ldots, n\} \wedge x'_{i_{r-1}} \leq x_j < x'_{i_r}\}| \tag{44}$$

$$\leq |\{x_j \mid j \in \{1, 2, \ldots, n\} \wedge x'_{i_{r-1}} < x_j < x'_{i_r}\}| + 1 \tag{45}$$

$$\leq i_r - i_{r-1}. \tag{46}$$

Since $\mathbb{E}[i_r - i_{r-1}] = \frac{n}{m+1}$ and $m = \lfloor \alpha n \rfloor$, we have $\mathbb{E}[i_r - i_{r-1}] = \mathcal{O}(1)$. Thus, we have $\mathbb{E}[|\{v_j \mid j \in \{1, 2, \ldots, l\} \wedge p_j = r\}|] = \mathcal{O}(1)$.

From the above, we know $\mathbb{E}[\eta_i] = \sum_{r=p_i}^{p_i+2\varepsilon} \mathcal{O}(1) = \mathcal{O}(\varepsilon)$. Therefore, the expected time complexity of the modified Displacement Sort is:

$$\mathbb{E}\left[\mathcal{O}\left(l + \sum_{i=1}^{l} \log(\eta_i + 1)\right)\right] = \mathcal{O}(l) + \mathcal{O}\left(\sum_{i=1}^{l} \mathbb{E}[\log(\eta_i + 1)]\right) \tag{47}$$

$$\leq \mathcal{O}(l) + \mathcal{O}\left(\sum_{i=1}^{l} \log(\mathbb{E}[\eta_i] + 1)\right) \tag{48}$$

$$= \mathcal{O}(l) + \mathcal{O}\left(\sum_{i=1}^{l} \log(\varepsilon + 1)\right) \tag{49}$$

$$= \mathcal{O}(l + l \log(\varepsilon + 1)) \tag{50}$$

$$= \mathcal{O}(n + n \log(\varepsilon + 1)). \tag{51}$$

$\square$

**Proof of Theorem 4.4** Using the above lemmas, we now provide the proof of Theorem 4.4 for both cases where modifications (i) and (ii) are applied.

*Proof of Theorem 4.4 (i).* Let $S(n)$ be the time complexity of Index2Sort when modification (i) is applied. In this case, the complexities of the steps ①, ②, ③, ④, and ⑥ remain exactly the same as in Theorem 4.1. For the step ⑤, the time complexity of sorting $\boldsymbol{v}$ using the approximate rank query results as predictions is $\mathcal{O}(n + n \log(\varepsilon + 1))$, as shown in Theorem A.3. Therefore, $S(n)$ can be expressed recursively as follows:

$$S(n) = S(\alpha n) + \mathcal{O}(\alpha n C(\alpha n) + (1 - \alpha)n Q(\alpha n) + n + n \log(\varepsilon + 1)). \tag{52}$$

By applying mathematical induction in the same way as in the proof of Theorem 4.1, we conclude that $S(n) = \mathcal{O}(nC(n) + nQ(n) + n \log(\varepsilon + 1))$. $\square$

*Proof of Theorem 4.4 (ii).* Let $S(n)$ be the time complexity of Index2Sort when modification (ii) is applied. In this case, the complexities of the steps ①, ②, ③, ④, and ⑥ remain exactly the same as in Theorem 4.1.

For the step ⑤, the time complexity of performing the exponential search for each element is $\mathcal{O}((1 - \alpha)n \log(\varepsilon + 1))$, because the difference between the approximate rank query result and the true rank query result is at most $\varepsilon$. Additionally, the total time complexity of sorting each range bucket is $\mathcal{O}(n)$ from Theorem A.1.

Therefore, $S(n)$ can be expressed recursively in the same form as Equation (52). Consequently, by following the same steps as in the proof of Theorem 4.4 (i), we conclude that $S(n) = \mathcal{O}(nC(n) + nQ(n) + n\log(\varepsilon + 1))$. □

## A.3 Proof of Theorem 4.5

*Proof of Theorem 4.5.* Let $S(n)$ denote the worst-case time complexity of Index2Sort. Following an approach similar to the proof of Theorem 4.1, the time complexity of each step in the Index2Sort algorithm is as follows:

① The worst-case complexity of splitting the array (including the optional shuffle) is $\mathcal{O}(n)$, as in Theorem 4.1.

② Sorting $\boldsymbol{u}$ recursively takes $S(\alpha n)$ in the worst case.

③ Constructing the index on $\boldsymbol{u}$ has a worst-case complexity of $\mathcal{O}(\alpha nC(\alpha n))$.

④ Answering rank queries for all elements of $\boldsymbol{v}$ using the index has a worst-case complexity of $\mathcal{O}((1 - \alpha)nQ(\alpha n))$.

⑤ Sorting $\boldsymbol{v}$ with bucket sort using the rank query results has a worst-case complexity of $\mathcal{O}(R((1 - \alpha)n))$.

⑥ Merging $\boldsymbol{u}'$ and $\boldsymbol{v}'$ has a worst-case complexity of $\mathcal{O}(n)$.

In the step ⑤, one of the worst-case scenarios occurs when $\Omega((1-\alpha)n)$ elements are placed into a single range bucket. In this case, the time complexity of sorting that range bucket is $\Omega(R((1 - \alpha)n))$. This represents the worst-case scenario due to the superadditivity of $R(n)$.

Thus, $S(n)$ can be expressed recursively as follows:

$$S(n) = S(\alpha n) + \mathcal{O}(\alpha nC(\alpha n) + (1 - \alpha)nQ(\alpha n) + R((1 - \alpha)n)). \tag{53}$$

Using the superadditivity of $R(n)$ and following the same mathematical induction approach as in the proof of Theorem 4.1, we conclude that $S(n) = \mathcal{O}(nC(n) + nQ(n) + R(n))$. □

# B A Generalized Framework for "Algorithms with Predictors"

In this section, we outline an initial method for applying the techniques and theoretical frameworks of algorithms with predictions to problem settings where only the training and inference algorithms of the machine learning model are provided as an opaque box. Specifically, consider a task $T$ where, given a data sequence $\boldsymbol{x}$, the goal is to derive its corresponding ground truth $\boldsymbol{x}'$ ($\boldsymbol{x}$ and $\boldsymbol{x}'$ do not necessarily have the same number of elements). For the sorting problem, $\boldsymbol{x}$ represents the input array, and $\boldsymbol{x}'$ is the sorted version of $\boldsymbol{x}$. In this problem setting, assume the existence of the following algorithms:

- **Predictor Training Algorithm.** For sufficiently large $n$, given a task with $n$ elements and its ground truth, a "predictor" can be trained with a time complexity of $\mathcal{O}(nC(n))$. This predictor satisfies the following properties: given a task with $m$ elements, it can output predictions with time complexity of $\mathcal{O}(mQ(n, m))$, and the "error" of the predictions is at most $\varepsilon$.

- **Algorithm with Predictions.** For sufficiently large $n$ and any $\eta \geq 0$, given a task with $n$ elements and predictions for each element (with a maximum "error" of $\eta$), the task can be completed with time complexity of $\mathcal{O}(P(n, \eta))$.

- **Greedy Algorithm.** For any $n$, given a task with $n$ elements, the ground truth can be obtained in finite time.

---

**Algorithm 2** Algorithm-With-Predictors

---

1: **Algorithms:**
2:   $\mathcal{A}_c$: Predictor Training Algorithm.
3:   $\mathcal{A}_p$: Algorithm with Predictions.
4:   $\mathcal{A}_g$: Greedy Algorithm.
5:
6: **function** Algorithm-With-Predictors($\boldsymbol{x}$)
7:     $n \leftarrow |\boldsymbol{x}|$
8:     **if** $n < \tau$ **then**
9:         **return** $\mathcal{A}_g(\boldsymbol{x})$
10:    $\boldsymbol{u} \leftarrow \boldsymbol{x}[1 : \lfloor n/2 \rfloor]$              ▷ ①
11:    $\boldsymbol{u}' \leftarrow$ Algorithm-With-Predictors($\boldsymbol{u}$)     ▷ ②
12:    $\mathcal{I} \leftarrow \mathcal{A}_c(\boldsymbol{u}, \boldsymbol{u}')$                ▷ ③
13:    $\hat{\boldsymbol{p}} \leftarrow \mathcal{I}.\text{predict}(\boldsymbol{x})$             ▷ ④
14:    **return** $\mathcal{A}_p(\boldsymbol{x}, \hat{\boldsymbol{p}})$            ▷ ⑤

---

Here, the "predictor" does not necessarily need to utilize machine learning; a simpler structure is sufficient. Additionally, the "error" is assumed to be a scalar value defined by an appropriate metric for the specific problem. For example, in the sorting problem, we can define the "error" $\eta := \sum_{i=1}^{n} \log(\eta_i^\Delta + 2)$, where $\eta_i^\Delta$ denote the error between the actual sorted position and the predicted position of the $i$-th element. Under this definition, the time complexity of the Displacement Sort proposed in (Bai & Coester, 2023) is $\mathcal{O}(\eta)$. That is, in this case, $P(n, \eta) = \mathcal{O}(\eta)$.

Here, let the function $C(n)$ be a non-decreasing function of $n$, and let $Q(n, m)$ be a non-decreasing function of both $n$ and $m$. This assumption reflects the natural idea that as the number of data points increases, the computational cost per element for training or inference of the predictor also increases. Note that the computational cost itself does not necessarily need to be monotonic; the assumption of monotonicity applies only to the upper-bound expression.

Similarly, let the function $P(n, \eta)$ be a non-decreasing function of both $n$ and $\eta$. This implies that the time complexity of an algorithm with predictions increases with the number of input elements or with larger errors in the predictions, which is a natural assumption. Again, this monotonicity assumption applies only to the upper-bound expression and does not require the time complexity itself to be strictly monotonic.

Additionally, let $P(n, \eta)$ be a superadditive function with respect to $n$. Specifically, for any $n_1 \geq 0$, $n_2 \geq 0$, and $\eta \geq 0$, we have $P(n_1 + n_2, \eta) \geq P(n_1, \eta) + P(n_2, \eta)$. This assumption reflects the natural notion that the time complexity required to solve a task with $n$ elements is at least as large as the sum of the complexities required to solve two subproblems split from the original task. Again, superadditivity is assumed for the upper-bound expression, not necessarily for the time complexity itself.

Under these conditions, the following theorem holds:

> **Theorem B.1.** *For a task $T$ as defined above, suppose the three algorithms described earlier exist. Then, given data with $n$ elements (without any accompanying predictions or ground truth), there exists an algorithm that can derive the ground truth with a time complexity of $\mathcal{O}(nC(n) + nQ(n, n) + P(n, \varepsilon))$.*

*Proof of Theorem B.1.* Here, the proof is constructive. First, we define the algorithm and then provide proof of its time complexity guarantees.

We define the necessary notation. Let $\mathcal{A}_c$ denote the predictor training algorithm, $\mathcal{A}_p$ denote the algorithm with predictions, and $\mathcal{A}_g$ denote the greedy algorithm. The input data sequence is denoted as $\boldsymbol{x}$, containing $n$ elements.

The algorithm, referred to as *Algorithm-With-Predictors*, is fundamentally similar to Index2Sort. The pseudocode for Algorithm-With-Predictors is presented in Algorithm 2. When the number of elements $n$ in the input data sequence is less than a constant $\tau$, the algorithm uses $\mathcal{A}_g$ to obtain the ground truth. For cases where $n \geq \tau$, the algorithm proceeds as follows:

① Extract half of the data from the input sequence $\boldsymbol{x}$ to create a new sequence $\boldsymbol{u}$.

② Recursively call Algorithm-With-Predictors on $\boldsymbol{u}$ to obtain the ground truth $\boldsymbol{u}'$ for $\boldsymbol{u}$.

③ Call $\mathcal{A}_c$ with $\boldsymbol{u}$ and $\boldsymbol{u}'$ to train a "predictor."

④ Use the trained predictor to make predictions $\hat{\boldsymbol{p}}$ for $\boldsymbol{x}$.

⑤ Call $\mathcal{A}_p$ with $\boldsymbol{x}$ and $\hat{\boldsymbol{p}}$ to obtain the ground truth for $\boldsymbol{x}$.

Now, we give the theoretical guarantees on the time complexity of Algorithm-With-Predictors. Let the time complexity of Algorithm-With-Predictors be $S(n)$. We prove that $S(n) = \mathcal{O}(nC(n) + nQ(n,n) + P(n,\varepsilon))$. The time complexity of each step in the Algorithm-With-Predictors algorithm is as follows:

① Extracting half of the data runs in $\mathcal{O}(n)$.

② Recursively calling Algorithm-With-Predictors on $\boldsymbol{u}$ takes $S(n/2)$.

③ Training a predictor with $\boldsymbol{u}$ and $\boldsymbol{u}'$ runs in $\mathcal{O}((n/2)C(n/2))$ (based on the assumptions for $\mathcal{A}_c$).

④ Making predictions $\hat{\boldsymbol{p}}$ for $\boldsymbol{x}$ using the predictor runs in $\mathcal{O}(nQ(n/2,n))$ (based on the assumptions for $\mathcal{A}_c$).

⑤ Deriving the ground truth for $\boldsymbol{x}$ using $\boldsymbol{x}$ and $\hat{\boldsymbol{p}}$ runs in $\mathcal{O}(P(n,\varepsilon))$ (based on the assumptions for $\mathcal{A}_p$).

For the complexity guarantee in ⑤, the fact is used that the "error" in the predictions obtained by ④ is at most $\varepsilon$ from the assumptions for $\mathcal{A}_c$. Thus, $S(n)$ can be expressed recursively as follows:

$$S(n) = S(n/2) + \mathcal{O}((n/2)C(n/2) + nQ(n/2,n) + P(n,\varepsilon)). \tag{54}$$

Using the superadditivity of $P$ and following a similar mathematical induction argument as in the proof of Theorem 4.1, we conclude $S(n) = \mathcal{O}(nC(n) + nQ(n,n) + P(n,\varepsilon))$. $\qquad\square$

## C  Theoretical Guarantee for ESPC-Index

Here, we provide proof for the following time complexity guarantees of ESPC-index, which were not explicitly mentioned in the original paper (Croquevielle et al., 2025).

> **Theorem C.1.** *The ESPC-index, with appropriately adjusted parameters, satisfies $C(n) = 1$ and $Q(n) = \log\log n$ under the assumption that $\boldsymbol{D} \overset{\text{iid}}{\sim} \chi \in \mathfrak{X}_C$ and queries are independently drawn from the same distribution $\chi$. That is, the expected time complexity for construction is $\mathcal{O}(n)$, and the expected time complexity for a single rank query is $\mathcal{O}(\log\log n)$.*

*Proof of Theorem C.1.* The proof follows a similar approach to the proof of Theorem 10 in the original ESPC-index paper (Croquevielle et al., 2025). In Theorem 10, it is shown that for an ESPC-index with parameter $K$ (representing the number of "subintervals" in the ESPC-index), the expected time complexity for construction is $\mathcal{O}(n + K)$, and the expected time complexity for a single rank query is $\mathcal{O}\left(\log \frac{n \log n}{K}\right)$.

In Theorem 10 of (Croquevielle et al., 2025), the time complexities are analyzed for $K = n \log n$. If we instead consider $K = n$, the expected time complexity for construction becomes $\mathcal{O}(n)$, and the expected time complexity for a single rank query becomes $\mathcal{O}(\log\log n)$. $\qquad\square$

## D   Details of the Derived Complexity Guarantees of Index2Sort

In this appendix, we provide the details omitted from the main text for each of the derived complexity guarantees summarized in Table 1. For each instantiation, we identify the relevant construction cost $C(n)$ and query cost $Q(n)$ of the underlying index, and then apply the corresponding theorem from Section 4.2 to obtain the resulting complexity of Index2Sort.

**Trivial but Revealing Case.**   As the most trivial case, consider answering rank queries using binary search without constructing an index. Here, our Index2Sort closely resembles the existing "Index Sort" (Gurram & Gera, 2011) (note that "Index" here refers to array indices, not an index data structure). However, "Index Sort" does not provide any time complexity guarantees. Since $C(n) = 0$ and $Q(n) = \log n$, Theorem 4.1 shows that its time complexity is $\mathcal{O}(n \log n)$, which is a novel observation. For classical index data structures, such as B-tree, where $C(n) = Q(n) = \log n$, Theorem 4.1 implies that the time complexity of Index2Sort using this index is also $\mathcal{O}(n \log n)$.

**Structure-Agnostic Proof of $\mathcal{O}(n \log \log n)$ Complexity.**   Next, we present the computational guarantees of Index2Sort with learned indexes. The learned index of (Zeighami & Shahabi, 2023) assumes data points are sampled i.i.d. from $\chi \in \mathfrak{X}_{\rho_1, \rho_2, \mathcal{K}}$ and achieves $C(n) = Q(n) = \log \log n$. By Theorem 4.2, the time complexity of Index2Sort with this learned index is $\mathcal{O}(n \log \log n)$. This is equivalent to the guarantees in (Sato & Matsui, 2025; Zeighami & Shahabi, 2024), but our Index2Sort achieves the same result without requiring any observation of the internal structure of the learned index, making both the algorithm and its time complexity guarantees more intuitive.

**From $\mathcal{O}(n \log \log n)$ to $\mathcal{O}(n)$.**   Index2Sort using ESPC-index (Croquevielle et al., 2025) provides stronger guarantees than prior learned sorts. When the data and queries are independently sampled from $\chi \in \mathfrak{X}_{\rho_f, \mathcal{K}}$, ESPC-index achieves $C(n) = Q(n) = 1$. By Theorem 4.2, Index2Sort therefore runs in expected $\mathcal{O}(n)$ time under $\chi \in \mathfrak{X}_{\rho_f, \mathcal{K}}$. This is a **tighter** guarantee under **weaker** assumptions than prior learned sorts (Sato & Matsui, 2025; Zeighami & Shahabi, 2024), which achieve $\mathcal{O}(n \log \log n)$ under $\chi \in \mathfrak{X}_{\rho_1, \rho_2, \mathcal{K}}$.

$\mathcal{O}(n \log \log n)$ **under the Weakest Assumptions.**   Moreover, using ESPC allows Index2Sort to obtain strong expected complexity guarantees even under very weak distributional assumptions. When the data and queries are independently sampled from $\chi \in \mathfrak{X}_C$, ESPC-index achieves $C(n) = 1$ and $Q(n) = \log \log n$ (this result is not mentioned in the original paper, but we prove it in Appendix C). Therefore, by Theorem 4.2, the expected time complexity of Index2Sort is $\mathcal{O}(n \log \log n)$ under $\chi \in \mathfrak{X}_C$. This is the first theoretical guarantee for learned sorts under the very weak distributional assumption $\chi \in \mathfrak{X}_C$.

**Complexity Guarantees under Distribution Drift.**   Finally, the learned index of (Zeighami & Shahabi, 2024) assumes $\boldsymbol{D} \sim \boldsymbol{\chi} \subset \mathfrak{X}_{\rho_1, \rho_2, \mathcal{K}}$ and $\Delta(\boldsymbol{\chi}) \leq \delta$, yielding $C(n, \delta) = Q(n, \delta) = \log \log n + \log(\delta n)$. Thus, by Theorem 4.3, Index2Sort runs in $\mathcal{O}(n \log \log n + n \log(\delta n))$ expected time. Although this complexity guarantee is not novel because the index also supports insertion, we include this case to illustrate how Index2Sort inherits guarantees even under distribution shift. Future static learned indexes with theoretical guarantees under distribution shifts could be seamlessly incorporated into our framework in the same manner.

## E   Index2Sort in External Memory

In this appendix, we discuss Index2Sort in the external-memory setting. We refer to this variant as EM-Index2Sort. We prove that, with suitable implementation choices and parameter settings, EM-Index2Sort matches the optimal external-memory sorting I/O bound up to a constant factor (Theorem E.1).

We consider the standard external-memory model. The input consists of $N$ items stored on disk. The internal memory can hold $M$ items, and one I/O transfers one block of $B$ consecutive items between disk and internal memory. Following the convention in the external-memory literature, this appendix uses notation that differs from that in the other sections. In this model, the average-case and worst-case number of I/Os required for

sorting $N$ data items is $\text{Sort}_{M,B}(N) \coloneqq \Theta\left(\frac{N}{B} \log_{M/B} \frac{N}{B}\right)$ (Vitter, 2008, Theorem 5.1); see also Aggarwal & Vitter (1988).

We assume that the index used by EM-Index2Sort can be kept in internal memory during the distribution step. More precisely, we reserve a constant fraction $cM$ of internal memory for the index, where $c < 1$, and use the remaining $M' \coloneqq (1-c)M$ as the effective internal memory available to the rest of the algorithm. Our assumption is that, when the index is built on $S \coloneqq \Theta\left(\min\left\{\sqrt{\frac{M'}{B}}, \frac{N}{M'}\right\}\right)$ elements, its entire representation fits within the reserved $cM$ space, and its rank queries can be answered without incurring any I/Os. The role of this choice of $S$ will be explained in the next section.

We note that this assumption is mild for many index structures. For example, standard indexes such as B-trees and learned indexes such as the PGM-index use space near-linear in the number of indexed elements, possibly up to logarithmic factors. Thus, since $S = \mathcal{O}(\sqrt{\frac{M'}{B}})$, the index built on the $S - 1$ splitters has size at most $\tilde{\mathcal{O}}(\sqrt{\frac{M'}{B}})$, which fits in a small reserved portion of internal memory under the usual external-memory parameter regimes. Moreover, rank queries typically require only a constant or logarithmic amount of additional workspace. Thus, once the index representation is resident in internal memory, rank queries incur no further I/Os.

### E.1 Designing EM-Index2Sort

We now describe the algorithm of EM-Index2Sort. EM-Index2Sort is similar to the in-memory version of Index2Sort, but differs in three main aspects: how the elements used to build the index are selected, how the splitter elements are sorted, and how each bucket is sorted.

First, EM-Index2Sort selects $S - 1$ splitter elements. In the in-memory setting, Index2Sort randomly selects a number of elements proportional to the input size. In contrast, in the external-memory setting, we set $S = \Theta\left(\min\left\{\sqrt{\frac{M'}{B}}, \frac{N}{M'}\right\}\right)$, and select $S - 1$ splitter elements in the same spirit as external-memory distribution sort (Aggarwal & Vitter, 1988). The splitter elements are chosen so that the subsequent bucketing step produces buckets of nearly equal size. This choice is important because, in the external-memory setting, building an index on a number of elements proportional to the input size would create too many buckets. If the number of buckets is too large, we cannot keep one output buffer per bucket in internal memory, and the distribution step may incur excessive I/Os. To avoid this issue, we use a much smaller number of buckets and choose $S$ as above, following the standard choice in external-memory distribution sort; see, e.g., (Vitter, 2008).

Second, we sort the selected splitter elements using an efficient in-memory sorting algorithm. This differs from the in-memory version of Index2Sort, where the selected elements are sorted recursively using Index2Sort itself. Since $S$ is sufficiently small, this sorting step can be performed in internal memory without additional asymptotic I/O cost. Therefore, we simply sort the splitter elements using an efficient in-memory sorting algorithm.

Next, as in the original in-memory version of Index2Sort, we build an index on the sorted splitter elements. By assumption, this index can be kept in internal memory and occupies at most $cM$ space. We then use the index to obtain the rank of each input element among the splitter elements, and assign the element to the appropriate one of the $S$ buckets according to this rank. Because $S = \Theta\left(\min\left\{\sqrt{\frac{M'}{B}}, \frac{N}{M'}\right\}\right)$, the buckets can be distributed efficiently: we can maintain one output buffer per bucket in internal memory, and the distribution step incurs only $\mathcal{O}(N/B)$ I/Os.

Finally, we sort each bucket and concatenate the sorted buckets to obtain the final sorted output. In the original in-memory setting, each bucket could be sorted using any sorting algorithm with, for example, $\mathcal{O}(n^2)$ time complexity. In the external-memory setting, we instead sort each bucket using an external-memory sorting algorithm that achieves the optimal I/O bound, such as $M/B$-way merge sort (Aggarwal & Vitter, 1988; Vitter, 2001) or distribution sort (Aggarwal & Vitter, 1988). Alternatively, we may recursively apply

EM-Index2Sort to each bucket; this recursive implementation satisfies the same I/O guarantee. Either choice implements the bucket-sorting step in a manner consistent with the external-memory model.

### E.2 I/O Guarantees for EM-Index2Sort

We now state the I/O guarantee for EM-Index2Sort described above.

**Theorem E.1** (I/O complexity of EM-Index2Sort). *EM-Index2Sort performs $\mathcal{O}(\mathrm{Sort}_{M,B}(N))$ I/Os. Since sorting $N$ items requires $\Omega(\mathrm{Sort}_{M,B}(N))$ I/Os in the standard external-memory model, EM-Index2Sort matches the external-memory sorting I/O bound up to a constant factor.*

*Proof.* Since $c$ is a constant, we have $\mathrm{Sort}_{M',B}(N) = \Theta(\mathrm{Sort}_{M,B}(N))$. For $N \leq M'$, the input fits in internal memory and can be sorted with $\mathcal{O}(N/B)$ I/Os. For $N > M'$, we analyze the I/O cost of each step:

- **Splitter Selection:** Using the splitter-selection method of Aggarwal & Vitter (1988), we can select $S-1$ splitter elements in $\mathcal{O}(N/B)$ I/Os. Let $N_i$ denote the size of the $i$-th bucket for $i = 1, 2, \ldots, S$. The selected splitters guarantee that $\frac{N}{2S} \leq N_i \leq \frac{3N}{2S}$ for every $i$.

- **Index Building and Distribution:** The index fits in internal memory (0 I/Os). Since $S = \mathcal{O}(M'/B)$, we can maintain one output buffer per bucket in internal memory. Distributing all items to $S$ buckets via a sequential scan and buffered writes takes $\mathcal{O}(N/B)$ I/Os.

- **Bucket Sorting:** Sorting all $S$ buckets using an optimal external-memory sorting algorithm requires $\sum_{i=1}^{S} \mathcal{O}(\mathrm{Sort}_{M',B}(N_i))$ I/Os. If EM-Index2Sort is used recursively instead, the same bound follows by induction on the subproblem size. Since $N_i \leq \frac{3N}{2S}$ and $\sum_{i=1}^{S} N_i = N$, this I/O count is at most $\mathcal{O}\left(\frac{N}{B} \log_{M'/B} \frac{3N}{2SB}\right)$. By the choice $S = \Theta\left(\min\left\{\sqrt{\frac{M'}{B}}, \frac{N}{M'}\right\}\right)$, this is bounded by $\mathcal{O}\left(\frac{N}{B} \log_{M'/B} \frac{N}{B}\right) = \mathcal{O}(\mathrm{Sort}_{M',B}(N))$.

- **Concatenation:** Sequentially writing the concatenated sorted buckets takes $\mathcal{O}(N/B)$ I/Os.

Summing these costs, the total I/O complexity is $\mathcal{O}(\mathrm{Sort}_{M',B}(N)) = \mathcal{O}(\mathrm{Sort}_{M,B}(N))$. Since sorting $N$ items requires $\Omega(\mathrm{Sort}_{M,B}(N))$ I/Os, EM-Index2Sort matches the optimal lower bound up to a constant factor. $\square$

This result should be interpreted differently from the in-memory guarantees. In internal memory, the main idea of learned sorting is that accurate rank predictions can reduce the computational cost of sorting. In external memory, however, the dominant bottleneck is data movement: even with accurate rank predictions, the items must still be moved and laid out on disk in sorted order. Thus, EM-Index2Sort does not bypass the $\mathrm{Sort}_{M,B}(N)$ lower bound. Rather, the theorem shows that the index-based distribution idea can be used in external memory without asymptotically worsening the optimal sorting I/O complexity.

## F Full Sorting-Time Comparison

In this appendix, we provide the full sorting-time comparison omitted from the main text. In addition to the methods reported in Section 5.3, we include Radix Sort, Index Sort (Gurram & Gera, 2011), `boost::sort::spreadsort::float_sort` (Ross, 2002), Balanced Learned Sort (BLS), Unbalanced Learned Sort (ULS), and PCF Learned Sort (Sato & Matsui, 2025). We also evaluate additional Index2Sort variants built on the PGM-index and a B-tree. All experimental settings are the same as in Section 5.3: we set $\alpha = 1/32$, use approximate ranks with $\varepsilon = 64$, and use the same hybrid post-processing scheme in step ⑤.

Figure 6 shows the full sorting-time comparison. The additional results provide two observations that complement the main-text discussion. First, Index Sort exhibits expected $\mathcal{O}(n \log n)$-type growth across all tested distributions. This is consistent with the fact that, unlike Index2Sort with the ESPC index, Index Sort

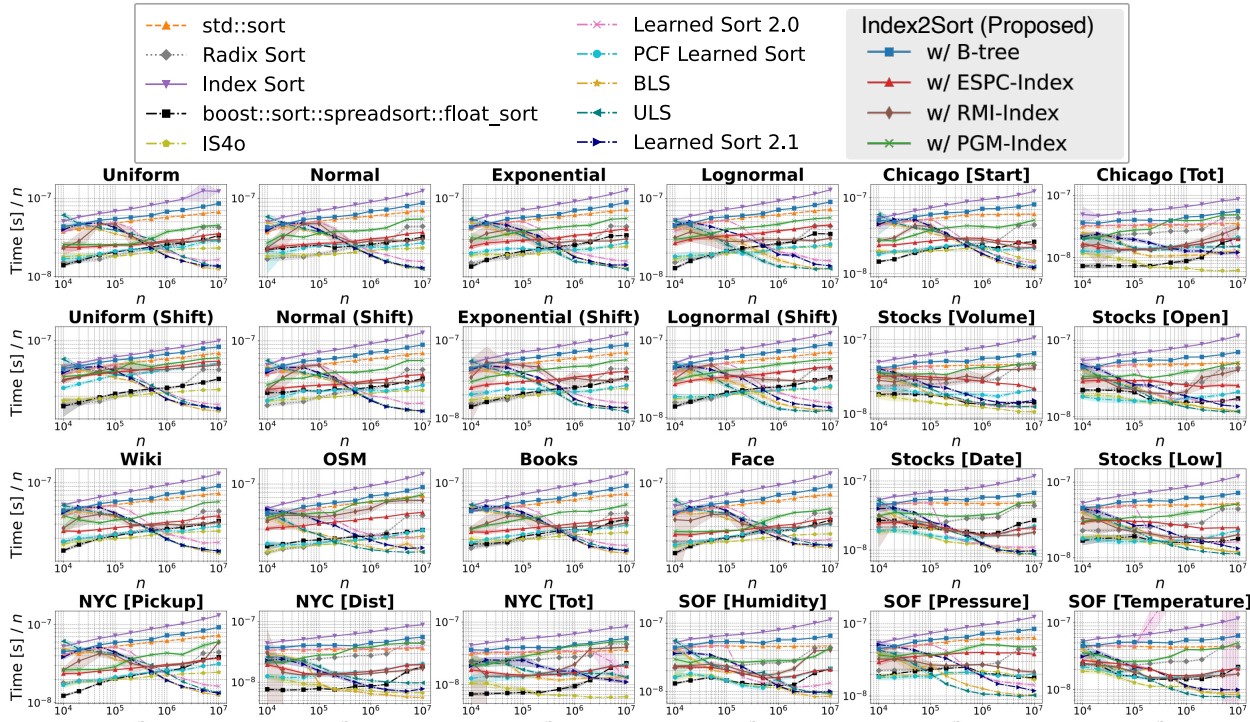

Figure 6: Sorting time vs. input length $n$.

does not exploit distributional structure to obtain sub-$n \log n$ expected running time. Second, Index2Sort with either a PGM-index or a B-tree also shows expected $\mathcal{O}(n \log n)$-type behavior in our experiments. This matches the guarantees obtained by instantiating Index2Sort with these indexes.

The additional Index2Sort variants further illustrate that practical performance depends on the underlying index, even when asymptotic bounds are similar. For example, although the PGM-index and B-tree have comparable worst-case asymptotic guarantees in this setting, the PGM-index is often faster in practice. The corresponding Index2Sort variant inherits this advantage and is generally faster than the B-tree-based variant.

The remaining learned sorting baselines, BLS and ULS, are highly optimized implementations and are often faster than our current ESPC-based Index2Sort implementation. This is consistent with the main-text observation for Learned Sort 2.1 and IS$^4$o: our implementation is intended to validate the theoretical abstraction and guarantees of Index2Sort, rather than to provide a cache-optimized sorting library. PCF Learned Sort also follows the expected sub-$n \log n$ trend on distributions covered by its assumptions, but it is less general than Index2Sort in the sense that Index2Sort can be instantiated with any static index supporting construction and rank queries.

Overall, the full comparison does not change the conclusions of Section 5.3. Rather, it confirms that the omitted baselines and index choices behave consistently with their corresponding theoretical guarantees and implementation characteristics.

# G   Breakdown of Sorting Time under Different Sampling Ratios

In this appendix, we analyze how the sorting time changes with the sampling ratio $\alpha$. While the main text reports the performance of Index2Sort with $\alpha = 1/32$, here we present a more detailed breakdown for $\alpha \in \{1/128, 1/64, 1/32, 1/16, 1/8, 1/4, 1/2\}$. For each value of $\alpha$, we fix the input size to $n = 10^7$ and decompose the total sorting time into three components: index construction, rank queries, and range-bucket sorting. The remaining time, including measurement overhead and other miscellaneous overheads, is reported

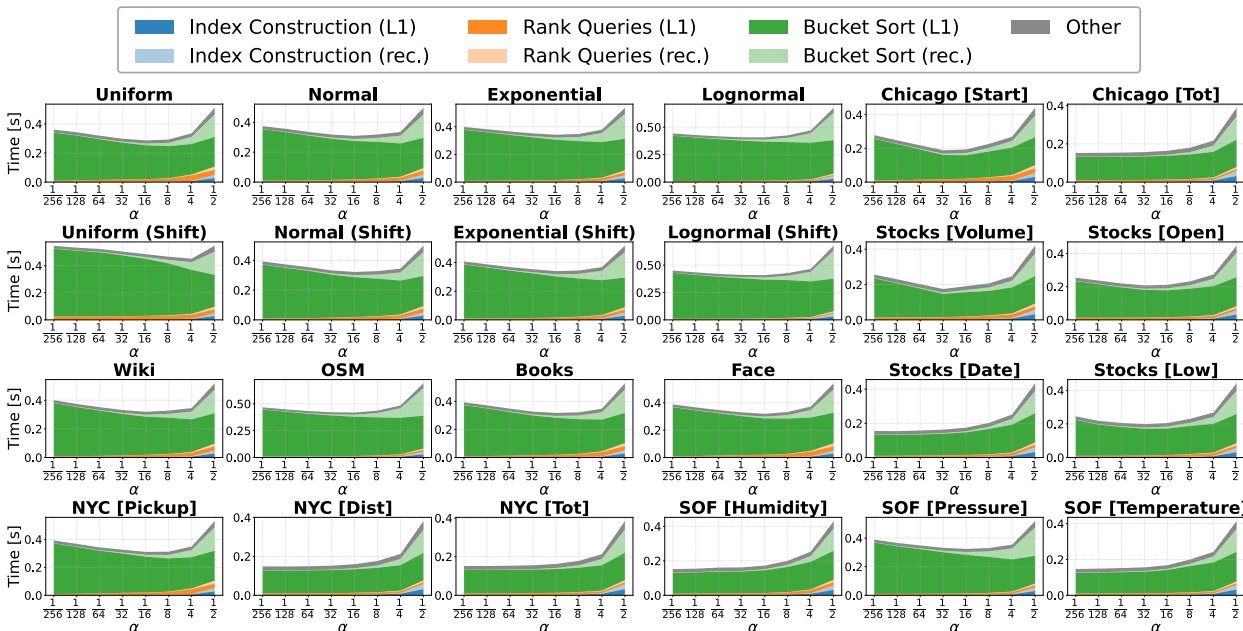

Figure 7: Breakdown of sorting time under different sampling ratios for Index2Sort with ESPC index.

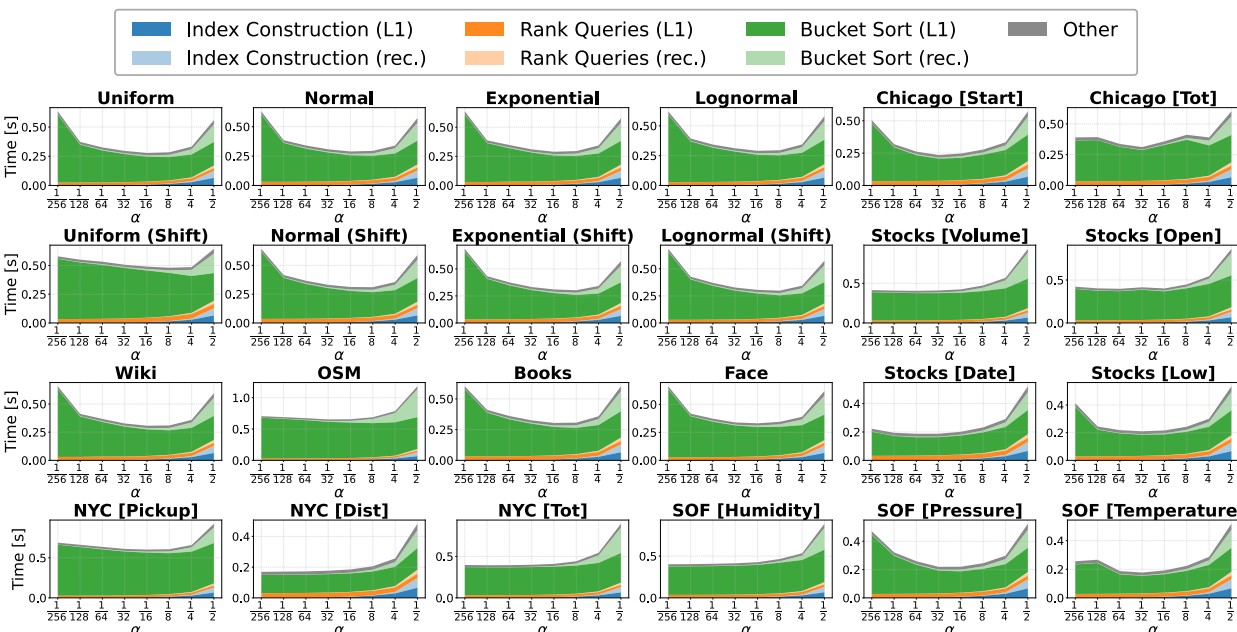

Figure 8: Breakdown of sorting time under different sampling ratios for Index2Sort with RMI index.

as Others. For each component, we further separate the time spent in the first recursion level (L1) from the time spent in subsequent recursive calls (rec.).

Figure 7 and Figure 8 show the breakdown of sorting time under different sampling ratios for Index2Sort with the ESPC index and the RMI index, respectively. In both cases, bucket sorting accounts for the largest portion of the running time. In particular, bucket sorting consistently accounts for more than 60% of the total sorting time. On the other hand, as $\alpha$ increases, the time spent on index construction also increases, because the constructed index becomes larger. This effect is particularly noticeable for the RMI index, where

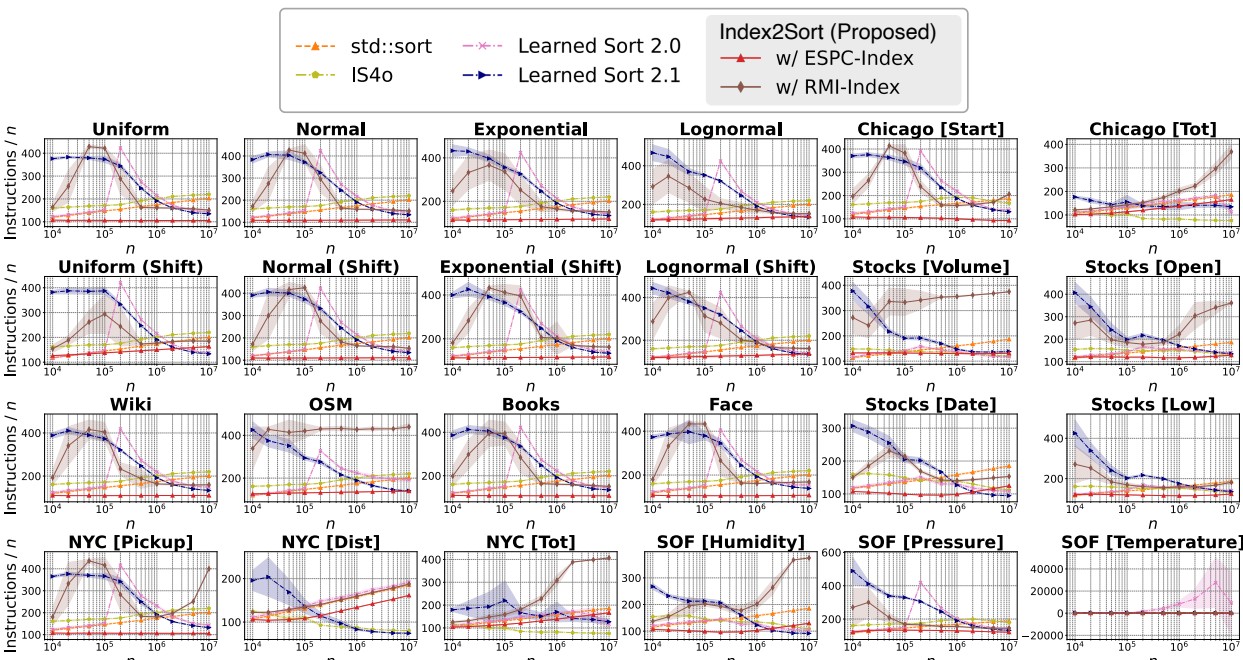

Figure 9: Instruction count per element for Index2Sort and the baselines. Index2Sort with the ESPC-index keeps the instruction count nearly constant across many datasets.

index construction accounts for up to 30% of the total running time when $\alpha$ is large. Thus, particularly when $\alpha$ is moderately large, it becomes important to choose an index that can be constructed efficiently.

Smaller values of $\alpha$ reduce the time spent on index construction and rank queries, since the constructed index is smaller. On the other hand, when $\alpha$ is too small, the time spent on bucket sorting increases, because each bucket contains more elements and sorting individual buckets becomes more expensive. Conversely, bucket sorting time can also increase when $\alpha$ is large. This is because a larger $\alpha$ increases the number of elements passed to recursive calls, resulting in deeper recursion. Although the bucket-sorting time at the first recursion level decreases as $\alpha$ increases, the time spent in subsequent recursive calls becomes larger. Overall, the total sorting time is minimized when $\alpha$ is roughly between 1/32 and 1/8.

## H  Instruction Count Measurement

In this appendix, we report the instruction counts of Index2Sort and the baseline sorting algorithms. The goal of this measurement is to complement the wall-clock-time results with a metric that is less directly affected by factors such as memory latency, cache behavior, and OS scheduling noise. Although instruction counts still depend on the compiler, optimization flags, and target architecture, they provide a useful proxy for comparing the amount of work performed by each method.

We measured instruction counts using the Linux `perf_event_open` interface. For each run, we counted the number of instructions executed during the sorting procedure using the hardware counter `PERF_COUNT_HW_INSTRUCTIONS`. Each experiment was repeated ten times with different input-generation seeds. The figure reports the average instruction count per element over these runs, with shaded regions indicating one standard deviation.

The results are shown in Figure 9. For most datasets and input sizes, Index2Sort with the ESPC-index executes the fewest instructions among all methods. In particular, its instruction count per element remains nearly constant for many datasets as $n$ increases, indicating that the total instruction count grows approximately linearly in $n$. This behavior is consistent with our expected-time guarantee for inputs satisfying the assumptions of our analysis.

On the other hand, for some datasets, such as NYC [Dist] and NYC [Tot], the instruction count per element increases with $n$. This suggests that the total instruction count grows closer to $n \log n$ on these datasets, rather than linearly. A possible explanation is that these datasets do not satisfy the condition $\chi \in \mathfrak{X}_{\rho_f, \mathcal{K}}$ required for the expected linear-work behavior. Nevertheless, even in these cases, Index2Sort with the ESPC-index executes the same number of instructions as, or fewer instructions than, STD::SORT in our experiments. This observation is consistent with our worst-case analysis: when the learned index does not provide sufficiently favorable bucket sizes, Index2Sort can still fall back to the comparison-sorting cost on the buckets, leading to $n \log n$-type behavior rather than worse asymptotic growth.

