# OpenReview forum: "Index2Sort: Sorting Algorithm Using Static Index Structure"
_TMLR — Accepted by TMLR_

### Review · Reviewer_GWca · 2026-04-02

**Summary Of Contributions:**

The paper introduces index2sort, an algorithm that decorates a rank query data structure to produce a sorting algorithm in a very natural and intuitive fashion by sampling from the input to produce a rank query data structure (index) which then essentially has the sampled elements act as pivots for bucket sort. The structures involved in the decorator are sufficiently fast and simple that the bottleneck becomes the index. So an $O(nC(n))$ construction $O(Q(n))$ query data structure becomes an $O(n(C(n)+Q(n)))$ sorting algorithm.
Some natural experiments are then performed that support the theoretical runtimes.
A great strength of the paper is that I couldn’t help myself but implementing the algorithm and playing around with the parameters, this speaks to both the presentation of and the algorithm itself being clean and interesting.

**Additional Comments:**

I think this is a good clean result overall.

**Audience:**

Yes

**Audience Explanation:**

Sorting is a central computer science problem and as the authors point out, learning the distribution of your input data can increase sorting performance so expanding this link between classical computer science and machine learning is great.

**Broader Impact Concerns:**

No broader impact concerns.

**Claims And Evidence:**

Yes

**Claims Explanation:**

The proofs and experiments are well presented and easy to follow.

**Requested Changes:**

My main concern is this:
The paper claims that an $\Omega(n^2)$ sorting algorithm suffices in the buckets, this is clearly only true for large m, say m=10 then pidgeon hole makes sorting one of the buckets Omega(n^2) which dominates reasonable indexes. For m = n//2 this is MAYBE fine, but I have a feeling that the birthday paradox should cause trouble, if it does not I would have liked to have seen that argument in the paper.
I would be surprised to see anyone use this framework with an $\Omega(n^2)$ sorting algorithm however, so it is not a major concern for the result.

The organization of the paper is a bit messy.
First of all, related work is section 5 which I find very odd.
The paper includes many applications of the framework, i.e. many indices where inserting the runtime functions C and Q in the main theorem trivially leads to optimal performance, again supported by experiments. I think that the paper spends too much time introducing these indices, the story could be much cleaner and about half the length if these were moved to appendix. Upon my first reading I thought all those distribution definitions would play a vital role, but then the high level argument is to simply use the C and Q from some other papers and insert in O(n(C(n)+Q(n))). It is great that the framework is useful, but it muddles the otherwise great story, in my opinion.

Index2sort has a sampling parameter m, a sample of size m is used to build the index. I think it would be very interesting to study what happens as m varies. The paper simply puts m = n//2 which in my personal (limited) experiments seems to be empirically suboptimal, setting m small means that the index will be a worse approximator, and buckets will be larger, on the other hand setting m large leads to limited bucket sort work but more merge work. In my experiments, counting comparisons in Python, with binary search as the index on uniformly random data small or large m were clear winners, with large m looking slightly superior.
Merging the u’ and v’ seems a bit odd, excuse my python but:
buckets = [[x] for x in u’]
for x in v:
     buckets[I(x)].append(x)
now we can flatten/concat buckets after sorting and there is no need to merge. I guess it doesn’t matter asymptotically, this leans more into my pivot view of your algorithm.

I am missing an external memory (memory M, blocksize B) analysis of index2Sort as it is very close to M/B-way mergeSort making it very natural to include some analysis here. If you use M/B-way mergeSort in the buckets and m=M/B then I think you will get a clean Sort(N) analysis, but there is some work to show that index performance is preserved.

Minor comments
The definition of sorting is lacking a $\pi$, $x_\pi(i)<=x_\pi(j)$ for all i,j s.t. $\pi(i)\leq\pi(j)$
If tau < m the algorithm goes into an infinite loop

---

> ### Author Response · Authors · 2026-05-16
>
> We sincerely appreciate the reviewer's insightful and constructive feedback. The comments have helped us significantly strengthen the manuscript. We have carefully revised the paper to address all the points raised, and the changes are highlighted in green in the updated manuscript PDF for the reviewer's convenience.
>
> ## Complexity of Bucket Sorting
>
> > My main concern is this: The paper claims that an O(n^2) sorting algorithm suffices in the buckets, this is clearly only true for large m, say m=10 then pidgeon hole makes sorting one of the buckets Omega(n^2) which dominates reasonable indexes. For m = n//2 this is MAYBE fine, but I have a feeling that the birthday paradox should cause trouble, if it does not I would have liked to have seen that argument in the paper. I would be surprised to see anyone use this framework with an  sorting algorithm however, so it is not a major concern for the result.
>
> We thank the reviewer for this insightful comment.
> We agree that using an $\mathcal{O}(n^2)$ sorting algorithm for the buckets is justified only when the number of buckets is sufficiently large, namely when $m = \Theta(n)$.
> If $m$ is small, then some bucket can indeed contain $\Theta(n)$ elements, and an $\mathcal{O}(n^2)$ bucket-sorting routine would indeed be problematic.
>
> When $m = \Theta(n)$, the birthday paradox implies that collisions occur with high probability.
> However, what matters for the running time is not whether collisions occur, but whether the total quadratic bucket-sorting cost remains small.
> Specifically, if $A_i$ denotes the number of elements in the $i$-th range bucket, we can show that the expected total cost satisfies $\mathbb{E}[\sum_i A_i^2] = \mathcal{O}(n)$.
> Thus, even an $\mathcal{O}(n^2)$ sorting algorithm inside each bucket contributes only linear expected total time under this setting.
>
> To clarify this point, we added a detailed discussion after Theorem 4.1 in Section 4.2, explaining why an $\mathcal{O}(n^2)$ bucket-sorting algorithm is sufficient when $m = \Theta(n)$.
>
> ## Paper Organization
>
> > First of all, related work is section 5 which I find very odd.
>
> We thank the reviewer for this helpful comment.
> We moved the Related Work section from Section 5 to Section 2.
>
> > The paper includes many applications of the framework, i.e. many indices where inserting the runtime functions C and Q in the main theorem trivially leads to optimal performance, again supported by experiments. I think that the paper spends too much time introducing these indices, the story could be much cleaner and about half the length if these were moved to appendix.
>
> We appreciate this constructive suggestion.
> We agree that the main theorem has a simple structure, and that applying it to each index mainly amounts to substituting the corresponding functions $C$ and $Q$.
> To improve the flow of the paper, we shortened the discussion in Section 4.3 on applying our theorem to different indexes, and moved the detailed descriptions to Appendix D.
>
> > Upon my first reading I thought all those distribution definitions would play a vital role, but then the high level argument is to simply use the C and Q from some other papers and insert in O(n(C(n)+Q(n))). It is great that the framework is useful, but it muddles the otherwise great story, in my opinion.
>
> We thank the reviewer for pointing out this potential source of confusion.
> We agree that the role of the distribution classes should be made clearer.
> Our main theorem treats the index as an opaque box, and therefore does not require us to revisit the detailed definitions of the distribution classes used in prior index results.
> Rather, these definitions are included for completeness, so that the assumptions behind the instantiated bounds are stated explicitly.
>
> To clarify this point, we revised the introduction of the distribution classes in Section 3, explicitly noting that the details of these classes are not used directly in the proof of our main framework theorem, but are included to make the assumptions of the instantiated learned-index guarantees self-contained.

---

> ### Author Response · Authors · 2026-05-16
>
> ## Ablation on $m$, i.e., the Fraction of Elements Used for Index Construction
>
> > Index2sort has a sampling parameter m, a sample of size m is used to build the index. I think it would be very interesting to study what happens as m varies.
>
> We thank the reviewer for suggesting this ablation study on the sampling parameter $m$.
> We conducted an additional ablation study on $m$, or equivalently on the sampling ratio $\alpha = m/n$, which controls the fraction of elements used to build the index.
> In this experiment, we varied $\alpha$ over $1/128, 1/64, 1/32, 1/16, 1/8, 1/4$, and $1/2$, and analyzed how the total sorting time and its components change.
>
> The results reveal a clear trade-off.
> When $\alpha$ is small, index construction and rank queries are cheaper because the constructed index is smaller.
> However, too small an $\alpha$ increases the bucket-sorting time, since each bucket contains more elements.
> Conversely, when $\alpha$ is large, the first-level bucket-sorting time decreases, but index construction becomes more expensive and more elements are passed to recursive calls, increasing the cost of subsequent recursion levels.
> Overall, the total sorting time is minimized when $\alpha$ is roughly between $1/32$ and $1/8$.
>
> We added these ablation results to Appendix G.
>
> ## Concatenation Instead of Merging
>
> > Merging the u' and v' seems a bit odd, excuse my python but: buckets = [[x] for x in u'] for x in v: buckets[I(x)].append(x) now we can flatten/concat buckets after sorting and there is no need to merge. I guess it doesn't matter asymptotically, this leans more into my pivot view of your algorithm.
>
> The reviewer is correct.
> In this paper, we adopt the merge-based description mainly for simplicity of exposition.
> As the reviewer points out, one can instead place the elements of $\mathbf{u}'$, the point buckets, and the range buckets in the appropriate order, sort the range buckets, and then simply concatenate them to obtain the final sorted output.
> This does not change the asymptotic complexity.
>
> We clarified this point at the end of Section 4.1.
>
> ## External-Memory Analysis
>
> > I am missing an external memory (memory M, blocksize B) analysis of index2Sort as it is very close to M/B-way mergeSort making it very natural to include some analysis here. If you use M/B-way mergeSort in the buckets and m=M/B then I think you will get a clean Sort(N) analysis, but there is some work to show that index performance is preserved.
>
> We thank the reviewer for pointing out the natural connection between Index2Sort and external-memory sorting.
> We added an extension of Index2Sort to the external-memory setting, together with its I/O analysis, in Appendix E.
> We refer to this variant as EM-Index2Sort.
>
> We prove that, with suitable implementation choices and parameter settings, EM-Index2Sort matches the optimal external-memory sorting I/O bound, namely $\mathrm{Sort}(N) = \mathcal{O}\left(\frac{N}{B}\log_{M/B}\frac{N}{B}\right)$, up to a constant factor (Theorem E.1).
>
> ## Minor Comment: Definition of Sorting
>
> > The definition of sorting is lacking a $\pi$, $x_{\pi}(i) \leq x_{\pi}(j)$ for all i,j s.t. $\pi(i) < \pi(j)$.
>
> We thank the reviewer for pointing out this imprecision.
> We corrected the definition of sorting in Section 3 to make it clearer and more precise.
>
> ## Minor Comment: Termination of the Recursion
>
> > If tau < m the algorithm goes into an infinite loop
>
> We thank the reviewer for raising this concern about termination.
> The algorithm does not enter an infinite loop, because the length of the array $\mathbf{u}$ passed to the next recursive call is always smaller than the length of the original array $\mathbf{x}$.
> Therefore, the recursion necessarily terminates.
>
> We clarified this point in Section 4.1.

---

> ### Comment · Reviewer_GWca · 2026-05-28
> **Small adjustment**
>
> I read through your changes and they look good, except for "This keeps each range bucket O(1) in size with high probability" on page 6.
> That is not true, see for instance https://scispace.com/pdf/balls-into-bins-a-simple-and-tight-analysis-3ud9nzzeh3.pdf and https://dl.acm.org/doi/10.1145/322248.322254 they show that the maximum size of a bucket is Theta(log n / log log n) with high probability, so while I agree that the expected total cost is linear, claiming that each individual bucket is constant size is wrong, even WHP.
>
> Edit: I forgot to add that I've never seen Sort_M(..) before, usually we write Sort(..) with the M and B being implicit. I would either change it to Sort_{M,B}(..) or to the standard Sort(..).

---

> ### Author Response · Authors · 2026-05-28
>
> We are grateful for the reviewer's reply and for the careful reading of our changes.
>
> We also appreciate the reviewer's comment on the statement on page 6. We agree that it was inaccurate, and we have revised it as follows:
>
> > This ensures that the expected sum of squared range-bucket sizes is $\mathcal{O}(n)$, and thus sorting all range buckets takes $\mathcal{O}(n)$ expected time.
>
> We also thank the reviewer for pointing out the notation for sorting in the external-memory model. We have changed the notation to $\mathrm{Sort}_{M,B}(\cdot)$.

---

### Review · Reviewer_4orV · 2026-04-13

**Summary Of Contributions:**

The authors introduce a reduction of sorting to indexing. The reduction preserves expected time complexity guarantees, which allow improvement upon the state-of-the-art theoretical guarantees (given various value distribution assumptions). Interestingly, it can be made resilient to approximate indexing with limited complexity overhead. Experiments confirm the complexity guarantees, but do not report time improvement upon state-of-the-art due to a lack of hardware optimization.

**Audience:**

Yes

**Audience Explanation:**

Data sorting is a fundamental problem in computer science, and the theoretical advances seem to be a fertile ground for overall progress in this area.

**Broader Impact Concerns:**

The paper does not provide a Broader Impact Statement, but the topic does not seem to require one.

**Claims And Evidence:**

Yes

**Claims Explanation:**

The paper leverages an elegant reduction of sorting to indexing. Albeit their careful analysis is not straightforward, the main guarantees feel quite intuitive.

**Requested Changes:**

- The dependency of  \mathfrak{X}_{\rho_1, \rho_2} and \mathfrak{X}_{\rho_f} on $\mathcal K$ should be highlighted. In particular, in Proposition 2.1, it should be made clear that $C$ depends on $\mathcal K$.
- Figure 5 is extremely hard to read. I'd suggest to only show the best of Index2Sort (I guess, with RMI-Index), and to limit as well the number of standard alternatives. The full graph could be provided in the Appendix.
- The comparisons with other hardware-optimized sorting algorithms is unflattering. Could the authors provide something more akin to the previous graphs, with a number of comparisons needed (or something else that is hardware-independent)?

---

> ### Author Response · Authors · 2026-05-16
>
> We are grateful to the reviewer for the careful reading and constructive comments, which have helped us substantially improve the clarity and quality of the manuscript. We have addressed all the points raised in detail. For the reviewer's convenience, the revised parts are highlighted in blue in the updated manuscript PDF.
>
> ## Notation for Distribution Classes
>
> > The dependency of `\mathfrak{X}_{\rho_1, \rho_2}` and `\mathfrak{X}_{\rho_f}` on $\mathcal{K}$ should be highlighted. In particular, in Proposition 2.1, it should be made clear that $C$ depends on $\mathcal{K}$.
>
> We thank the reviewer for highlighting this notation issue.
> We agree that the dependence of `\mathfrak{X}_{\rho_1, \rho_2}` and `\mathfrak{X}_{\rho_f}` on $\mathcal{K}$, as well as the dependence of the constant $C$ in Proposition 2.1 on $\mathcal{K}$, should be made explicit.
>
> To address this issue, we revised the notation for the distribution classes.
> Specifically, we changed `\mathfrak{X}_{\rho_1, \rho_2}` to `\mathfrak{X}_{\rho_1, \rho_2, \mathcal{K}}`, and `\mathfrak{X}_{\rho_f}` to `\mathfrak{X}_{\rho_f, \mathcal{K}}`.
> We also revised Proposition 2.1 to make clear that the constant $C$ depends on $\mathcal{K}$.
>
> ## Readability of Experimental Results
>
> > Figure 5 is extremely hard to read. I'd suggest to only show the best of Index2Sort (I guess, with RMI-Index), and to limit as well the number of standard alternatives. The full graph could be provided in the Appendix.
>
> We thank the reviewer for pointing out this important presentation issue.
> We reduced the number of Index2Sort variants and baseline methods shown in Figure 5 in the main text, and moved the full set of results to Appendix F.
> This improves the readability of the main figure and makes the key experimental message easier to see.
>
> ## Comparison Using a Less Hardware-Dependent Metric
>
> > The comparisons with other hardware-optimized sorting algorithms is unflattering. Could the authors provide something more akin to the previous graphs, with a number of comparisons needed (or something else that is hardware-independent)?
>
> We agree that a less hardware-dependent metric would strengthen the comparison.
> To complement the wall-clock-time comparison, we added an instruction-count measurement in Appendix H as a less hardware-dependent metric.
> Although instruction counts still depend on the compiler, optimization flags, and target architecture, they are less directly affected by memory latency, cache behavior, and OS scheduling noise than wall-clock time.
>
> The results show that Index2Sort with the ESPC-index executes the fewest instructions for most datasets and input sizes.
> Moreover, its instruction count per element remains nearly constant on many datasets, which is consistent with our expected linear-time guarantee under the assumed distributional conditions.
> Even for datasets where this trend does not hold, such as NYC [Dist] and NYC [Tot], Index2Sort still executes a comparable or smaller number of instructions than std::sort.
> This observation is consistent with our worst-case analysis.

---

### Review · Reviewer_zfu9 · 2026-04-28

**Summary Of Contributions:**

The paper proposes Index2Sort, a general framework that converts any static index data structure into a sorting algorithm by treating the index as an opaque box exposing only two operations: index construction and rank queries. The algorithm shuffles and splits the input into two halves $u$ and $v$, recursively sorts $u$, builds an index over the sorted $u'$, uses the index to bucket-sort $v$ via rank queries, and finally merges the two sorted halves.

The central theoretical contribution is the complexity-transfer theorem: if the underlying index admits construction in expected $O(nC(n))$ time and rank queries in expected $O(Q(n))$ time, then Index2Sort sorts in expected $O(nC(n) + nQ(n))$ time. This bound is established in three cases:
- distribution-free (Theorem 3.1), i.i.d. (Theorem 3.2)
- bounded distribution shift (Theorem 3.3)
- approximate-rank indexes (Theorem 3.4, adding an $n\log(\varepsilon+1)$ term) and to a worst-case bound (Theorem 3.5, adding an $R(n)$ term that typically resolves to $O(n \log n)$).

Empirically, the authors validate the theory on 24 artificial and real-world datasets, showing that the comparison count in the bucket-sort step scales linearly in $n$ and proportionally to $\log \varepsilon$ across multiple index types (B-tree, RMI, PGM, ESPC).

## Strengths

- The opaque-box abstraction is genuinely simple and inverts the standard dependency between sorting and indexing in a satisfying way. The framework is index-agnostic, which means future improvements in learned indexes automatically transfer to sorting.
- The expected $O(n)$ guantee under $\mathcal{X}_{\rho_f}$ is a strict impovement over the $O(n \log \log n)$ bounds of Sato & Matsui (2024) and Zeighami & Shahabi (2024), and is achieved under weaker distributional assumptions.
- The three main theorems share a single bucket-sort lema (Lemma A.1) and a unified recurrrence, making the analysis easy to follow and extend. The point-bucket construction handling duplicates is a tidy technical touch.

## Weaknesses

- The worst-case complexity $O(nC(n) + nQ(n) + R(n))$ leaves an dependency on the range-bucket sorter. It would strengthen the paper to either remove this term or argue more carefully why it is unavoidable.
- The "recursive half plus bucket sort" structure closely resembles classical samplesort and existing learned-sort pipelines; it looks like the the paper's novelty lies in the abstraction and analysis rather than in the algorithmic mechanics. I could be wrong here, but I think it's worth clarifying

**Audience:**

Yes

**Audience Explanation:**

The paper aligns with the categori of learned data structures and algorithms with predictions, and the broader program of using machine learning to obtain provable improvements over classical algorithmic baselines. Researchers working on learned indexes (e.g., the line of work from Kraska et al. (2018) through PGM-index, RMI, ALEX, and ESPC-index) would find the opaque-box reduction directly relevant.

Practitioners or system researchers might find the paper less interesting. The paper is upfront that wall-clock performance lags behind cache-tuned competitors, and a practitioner reading only Figure 5 might come away unimpressed. The interest is principally theoretical and conceptual, but within those bounds it is broad and well-aligned with what TMLR's audience reads.

**Broader Impact Concerns:**

I don't see any ethical issue from the manuscript.

**Claims And Evidence:**

Yes

**Claims Explanation:**

- The complexity-transfer theorems, I think, are the technical heart of the paper. The proofs in Appendix A are carefully written and modular: Lemma A.1 isolates the non-trivial bucket-sort analysis by adapting the classical Frazer & McKellar (1970) argument to the point-bucket setting, and the three main theorems then reduce to a shared recurrence solved by induction. The treatment of the i.i.d. and distribution-shift settings (Theorems 3.2 and 3.3) correctly identifies that the i.i.d. or shift-bounded property propagates from $x$ to the sub-arrays $u$ and $v$, which is exactly what is needed for the index's own guarantees to apply recursively.
- The headline corollary is the expected $O(n)$ sorting under $\mathcal{X}_{\rho_f}$, and it follows immediately from Theorem 3.2 once $C(n) = Q(n) = O(1)$ is established for the ESPC-index. The Appendix C argument re-tunes the ESPC parameter $K$ from $n \log n$ to $n$ and tracks the resulting expected bounds; this is a small but well-justified extension of Theorem 10 in the original ESPC paper.
- The somewhat weak part is the empirical eval. Figure 5 shows that ESPC-based Index2Sort scales substantially slower than $n \log n$ in practice, matching the predicted $O(n)$ on uniform inputs and $O(n \log \log n)$ on normal/exponential inputs. However, in absolute wall-clock time, ESPC-based Index2Sort loses on average to BLS, ULS, and Learned Sort 2.1.

**Requested Changes:**

- Does Index2Sort inherit amortized (vs. expected) bounds? Some indexes provide only amortized guarantees.
- The recursion rebuilds the index at each level (n/2, n/4, …). For learned indexes with expensive training (e.g., RMI), how much wall-clock time is spent on retraining at deeper levels?
- Quantitatively, how much slower is ESPC-based Index2Sort than the most optimized learned sorters (Learned Sort 2.1, BLS, ULS)?

---

> ### Author Response · Authors · 2026-05-16
>
> We sincerely thank the reviewer for the thoughtful and constructive feedback, which has greatly helped us improve the quality of our manuscript. We have carefully addressed all the points raised. In the updated manuscript PDF, the revised parts are highlighted in orange for the reviewer's convenience.
>
> ## Dependence of the Worst-Case Complexity on the Range-Bucket Sorter
>
> > The worst-case complexity $\mathcal{O}(nC(n) + nQ(n) + R(n))$ leaves an dependency on the range-bucket sorter. It would strengthen the paper to either remove this term or argue more carefully why it is unavoidable.
>
> We thank the reviewer for pointing out this important point.
> The dependence on $R(n)$ in the worst-case complexity is difficult to avoid because, in the worst case, all elements of $\mathbf{v}$ may be assigned to a single range bucket.
> In such a case, Index2Sort must explicitly sort a bucket of size $\Theta(n)$, and therefore the cost of the range-bucket sorting subroutine contributes the $R(n)$ term.
> Although the initial shuffle makes such a highly unbalanced bucket unlikely in the expected-time analysis, this possibility must still be accounted for in the worst-case analysis.
>
> To clarify this point, we added an explanation after Theorem 4.5 in Section 4.2, explaining why the worst-case bound retains the dependence on $R(n)$.
>
> ## Clarifying the Source of the Paper's Novelty
>
> > it looks like the the paper's novelty lies in the abstraction and analysis rather than in the algorithmic mechanics. I could be wrong here, but I think it's worth clarifying
>
> We appreciate the reviewer's thoughtful observation.
> We agree that the algorithmic structure of Index2Sort is intentionally simple, and that the main contribution of our paper lies in the abstraction of the index interface and the theoretical analysis, rather than in introducing a new bucketing mechanism itself.
>
> To make this explicit, we clarified this point in Section 1.
>
> ## Theoretical Analysis When the Index Provides Amortized Complexity Bounds
>
> > Does Index2Sort inherit amortized (vs. expected) bounds? Some indexes provide only amortized guarantees.
>
> We thank the reviewer for this thoughtful and constructive comment.
> Our complexity guarantees also extend to the case where the underlying index provides amortized bounds for rank queries.
> In particular, Index2Sort still obtains the same overall bound of $\mathcal{O}(nC(n) + nQ(n))$.
> This is because our analysis only requires that the total time spent on rank queries is bounded by $\mathcal{O}(nQ(n))$, rather than requiring every individual rank query to take $\mathcal{O}(Q(n))$ time.
>
> We clarified this point in the discussion following Theorem 4.1 in Section 4.2.
> This observation further broadens the applicability of our framework, and we thank the reviewer for raising it.
>
> ## Breakdown of Running Time
>
> > The recursion rebuilds the index at each level (n/2, n/4, ...). For learned indexes with expensive training (e.g., RMI), how much wall-clock time is spent on retraining at deeper levels?
>
> We thank the reviewer for this practical question about wall-clock time.
> To address this question, we conducted an additional experiment that analyzes the breakdown of the sorting time.
> In this experiment, we vary the sampling ratio $\alpha$, which controls the fraction of elements used to build the index, and decompose the total running time into index construction, rank queries, range-bucket sorting, and other overheads.
> For each component, we further separate the time spent at the first recursion level from the time spent in subsequent recursive calls.
>
> The main findings are as follows.
> First, most of the running time of Index2Sort is spent in the bucket-sorting routine: in our experiments, bucket sorting consistently accounts for more than 60\% of the total sorting time.
> Second, the time spent on index construction increases as $\alpha$ becomes larger, because the constructed index becomes larger and more elements are passed to recursive calls.
> This effect is particularly noticeable for learned indexes with relatively expensive training, such as RMI.
> For RMI, index construction accounts for up to about 30\% of the total running time when $\alpha$ is large.
> Third, the total sorting time is minimized when $\alpha$ is roughly between $1/32$ and $1/8$, reflecting the trade-off between reducing index-construction and rank-query costs and keeping bucket-sorting costs small.
>
> We added these experimental results to Appendix G.
>
> ## Quantitative Comparison with State-of-the-Art Sorters
>
> > Quantitatively, how much slower is ESPC-based Index2Sort than the most optimized learned sorters (Learned Sort 2.1, BLS, ULS)?
>
> We appreciate the reviewer's request for a quantitative evaluation of the slowdown compared to state-of-the-art learned sorters.
> Quantitatively, we found that ESPC-based Index2Sort is, on average, 2.0 times slower than the fastest sorting algorithm, with a maximum slowdown of 4.2 times.
> We added a discussion of this point in Section 5.3.

---

### Decision · Action_Editor_w4C8 · 2026-06-03

**Recommendation:** Accept as is

**Audience:**

Yes

**Audience Explanation:**

Yes, the reviewers all agreed this topic was of interest, and the results were novel, and move the understanding of this challenge forward.

**Claims And Evidence:**

Yes

**Claims Explanation:**

All of the reviewers were satisfied with the statement of the claims, and the proofs.  Thanks for working with reviewer comments to refine the few aspects that needed attention.